# Direct observation of translational activation by a ribonucleoprotein granule

Ruoyu Chen[1,2], William Stainier [1,3], Jeremy Dufourt [4,5], Mounia Lagha [4] & Ruth Lehmann [1,6]✉

Biomolecular condensates organize biochemical processes at the subcellular level and can provide spatiotemporal regulation within a cell. Among these, ribonucleoprotein (RNP) granules are storage hubs for translationally repressed mRNA. Whether RNP granules can also activate translation and how this could be achieved remains unclear. Here, using single-molecule imaging, we demonstrate that the germ cell-determining RNP granules in *Drosophila* embryos are sites for active translation of *nanos* mRNA. *Nanos* translation occurs preferentially at the germ granule surface with the 3′ UTR buried within the granule. Smaug, a cytosolic RNA-binding protein, represses *nanos* translation, which is relieved when Smaug is sequestered to the germ granule by the scaffold protein Oskar. Together, our findings uncover a molecular process by which RNP granules achieve localized protein synthesis through the compartmentalized loss of translational repression.

Biomolecular condensates compartmentalize the intracellular environment and biochemical processes to promote efficiency, achieve specificity and allow regulation at the spatiotemporal levels[1]. Ribonucleoprotein (RNP) granules are a type of condensate that serves as hubs of post-transcriptional regulation by localizing specific RNAs and RNA-binding proteins (RBPs)[2–4]. Most of the well-studied RNP granules, including stress granules[5], processing bodies[6,7] and neuronal transport granules[8], mainly assemble and store translationally repressed messenger RNA. The assembly of RNP granules can directly cause translational repression[9–11]. Translation resumes only when the stored mRNAs are released from the RNP granules or the granules undergo disassembly[12–16]. Conversely, it has been elusive but curious whether RNP granules can also activate the translation of stored mRNA[4]. Recently, evidence has emerged that condensation of specific RBP via liquid−liquid phase separation can activate the translation of their target mRNAs[17]. Several studies have reported that specific cytoplasmic RNP granules may serve as translation factories[18–21]. However, the role of RNP granules in translational activation, whether translation occurs on the granule and how it is regulated remain unclear.

In early *Drosophila* embryos, specialized RNP granules, called germ granules, located in the posterior cytoplasm (also known as germplasm; Fig. 1a) are essential for the formation of primordial germ cells (PGCs)[22–24]. Several maternally deposited mRNAs (for example, *nanos*, *gcl* and *pgc*) crucial for anterior–posterior patterning of the embryo and PGC specification are concentrated in the germplasm. Their translation is restricted to the germplasm even though these mRNAs are also present throughout the entire embryo[25–31]. It has therefore been proposed that germ granules may serve as the compartments for the translation of these mRNAs[21,28,31]. Thus, *Drosophila* germ granules serve as a model to gain insight into how RNP granules can control not only the storage but also the activation of translationally silenced RNAs.

Translational repression of *nanos* mRNA in the embryonic soma is mediated by the RBP Smaug, which binds to the *nanos* 3′ untranslated region (UTR) and recruits the translational repressors Cup, an eIF4E-binding protein, and the CCR4-NOT deadenylation complex[31–34]. Derepression of *nanos* translation at the posterior of the embryo has been attributed to germ granules, but the mechanism of derepression is not well understood[31,35]. The scaffold protein of germ granules, Oskar

[1]Whitehead Institute for Biomedical Research, Cambridge, MA, USA. [2]Vilcek Institute of Graduate Studies, NYU School of Medicine, New York, NY, USA. [3]Immunobiology Laboratory, The Francis Crick Institute, London, UK. [4]Institut de Génétique Moléculaire de Montpellier, University of Montpellier, Montpellier, France. [5]Institut de Recherche en Infectiologie de Montpellier, University of Montpellier, Montpellier, France. [6]Department of Biology, Massachusetts Institute of Technology, Cambridge, MA, USA. ✉e-mail: lehmann@wi.mit.edu

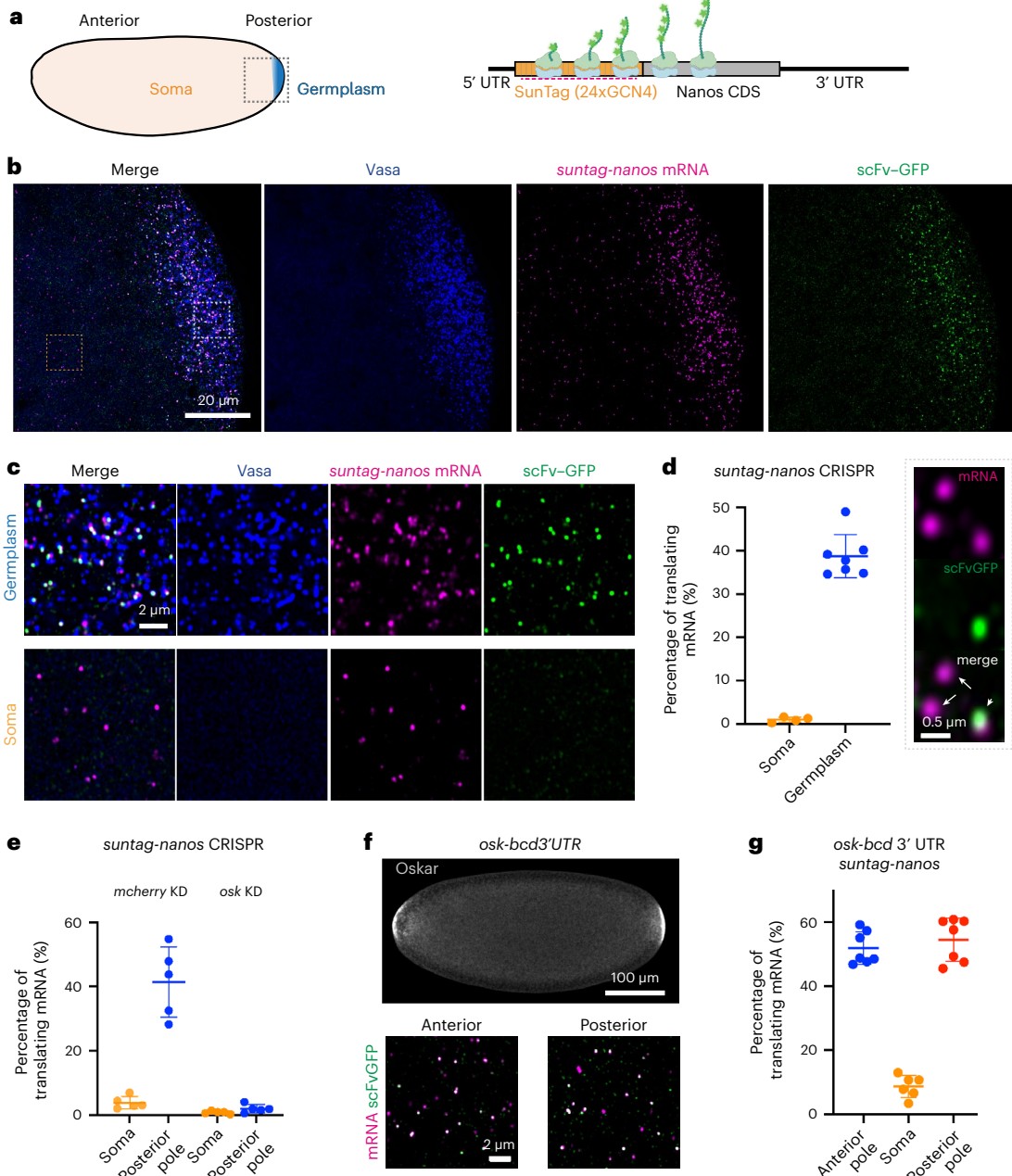

**Fig. 1 | Imaging translation of *nanos* mRNA in *Drosophila* embryos. a**, Left: schematic of a *Drosophila* embryo. Germplasm (blue) is located at the posterior pole of the embryo. The dashed square represents the region imaged by confocal microscopy and presented in **b**. Right: schematic of a translating *suntag-nanos* mRNA. A repetitive array of SunTag epitopes is added to the N-terminus of the *nanos* CDS. Nascent SunTag peptides are detected by scFv–GFP binding and *suntag* mRNA is detected by smFISH probes (magenta dashed line). **b**, A representative confocal image of the posterior pole of an embryo expressing Vasa–mApple (blue), *suntag-nanos* (mRNA stained by *suntag* smFISH probes, magenta), and scFv–GFP (green). Outlined regions in germplasm and soma are magnified and presented in **c**. **c**, Magnified images of germplasm and soma show the different translation activities in these two parts of the embryo.

**d**, Left: quantification of the percentage of translating mRNA in the soma (*n* = 4) and the germplasm (*n* = 7). Right: zoomed confocal images showing examples of a translating mRNA that co-localizes with scFv–GFP signal (arrowhead) and two non-translating mRNA that do not co-localize with scFv–GFP signal (arrows). **e**, Quantification of *suntag-nanos* mRNA translation in the soma and the posterior pole of embryos with *mCherry* knockdown (KD) or *osk* KD. *n* = 5 for all experiments. **f**, *Osk-bcd* 3′ UTR expression induces germplasm and translation of *suntag-nanos* mRNA at the anterior pole. Top: Oskar protein is immunostained with anti-Oskar antibody. Bottom: translation of *suntag-nanos* mRNA in native germplasm at the posterior and ectopic germplasm at the anterior, which are quantified in **g**. *n* = 7 (anterior), 6 (soma) and 7 (posterior). In **d**,**e**,**g**, the data are the mean ± s.d.; *n*, number of the embryos used for measurement.

(the short isoform), has been proposed to antagonize Smaug's function[31,32,36–39]. However, the mechanism has not been dissected in vivo owing to a lack of separation-of-function *oskar* alleles that specifically impede *nanos* translation without affecting germ granule assembly and RNA localization. In this study, we focused on the translational regulation of *nanos* mRNA by germ granules. By direct visualization

of *nanos* translation at the single-molecule level using the SunTag technique, we demonstrate that germ granules are the exact sites of *nanos* translation. Taking advantage of the quantitative nature of the SunTag system and a newly generated separation-of-function *oskar* allele, we dissected the mechanism of translational activation by germ granules in vivo at the molecular level.

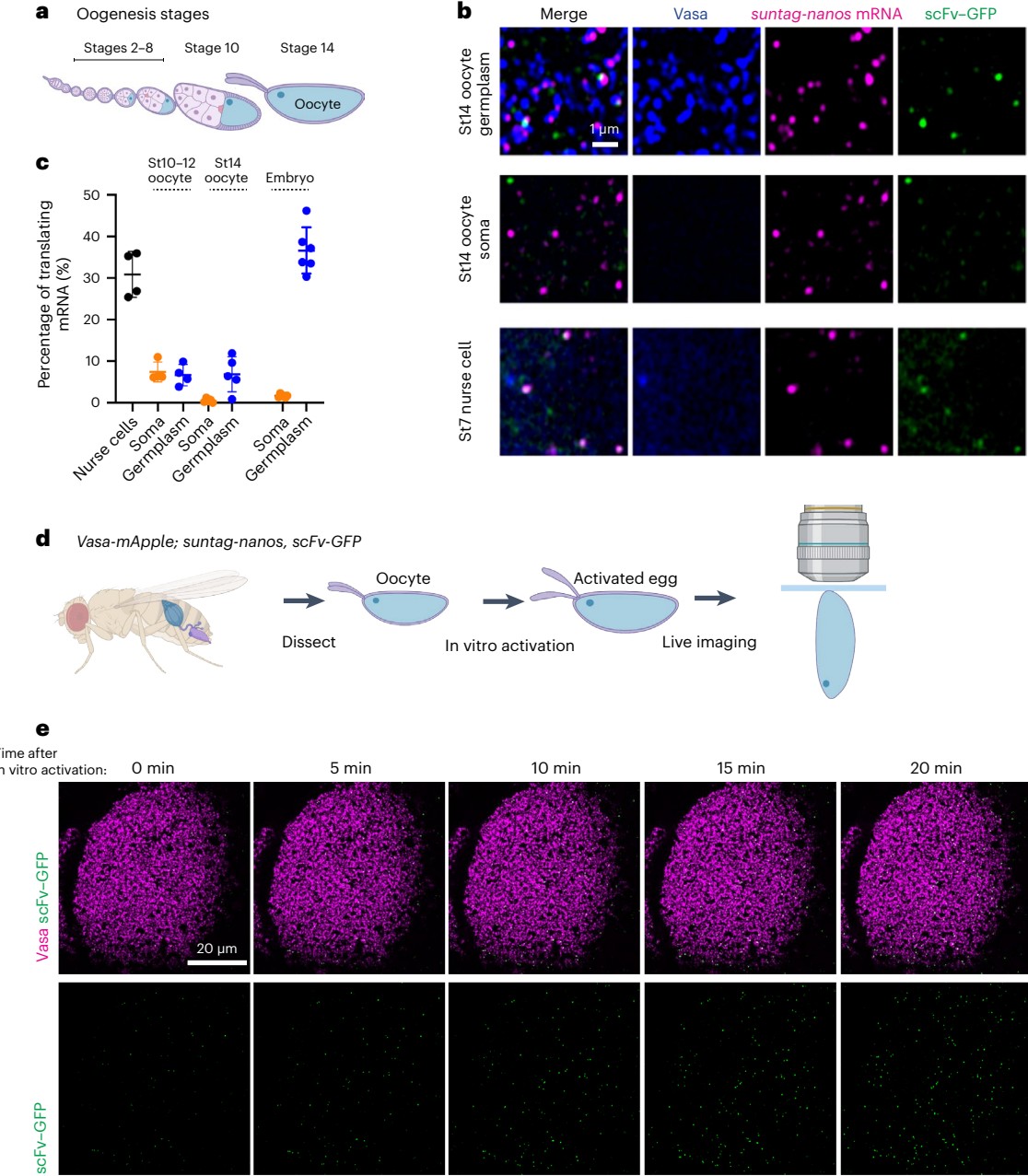

**Fig. 2 | Temporal regulation of localized translation in germplasm.**
**a**, Schematic of *Drosophila* oogenesis stages. **b**, Representative images of germplasm (top) and soma (middle) in stage (St)14 oocyte and cytoplasm of a stage 7 nurse cell (bottom) expressing *suntag-nanos* and scFv–GFP. Blue, Vasa; magenta, *suntag* smFISH; green, scFv–GFP. **c**, Translating fraction of *suntag-nanos* mRNA in stage 4–10 nurse cells (*n* = 4 egg chambers), soma and germplasm of stage 10–12 (developing) oocytes (*n* = 4 oocytes), stage 14 (mature) oocytes (*n* = 5 oocytes) and stage 1–2 embryos (*n* = 6 embryos). The data are the mean ± s.d. **d**, Protocol of in vitro activation of oocytes and live imaging. Mature oocytes are dissected from *Vasa-mApple*/+; *suntag-nanos*, *scFv-GFP*/+ flies and activated with 30% Robb's buffer (Methods). Activated eggs are mounted onto a coverslip and imaged by confocal microscopy. **e**, Representative time-lapse images of the germplasm of an activated egg with an increasing number of polysome (green foci). Germplasm is marked by Vasa–mApple (magenta), and SunTag is detected by endogenous scFv–GFP (green). The top shows the merged image, and the bottom shows scFv–GFP channel only. Schematics in **a** and **d** were generated with BioRender (https://www.biorender.com/).

## SunTag directly demonstrates localized translation of *nanos*

To investigate whether germ granules are compartments for active translation, we sought to visualize *nanos* translation in vivo at the single-molecule level. To this end, we used the SunTag system, whereby a repetitive array of a GCN4 epitope (SunTag) is appended to the CDS of the gene of interest and a green fluorescent protein (GFP)-fused single-chain antibody fragment (scFv–GFP) that binds the GCN4 epitope is co-expressed[40–44]. Observed by high-resolution light microscopy, the binding of scFv–GFP to GCN4 epitopes renders nascent peptides emerging from the polysomes as bright GFP foci. Translation of the SunTag can be correlated simultaneously with the corresponding mRNA signal visualized by single-molecule fluorescence in situ hybridization (smFISH; Fig. 1a). Using clustered regularly interspaced short palindromic repeats (CRISPR), we knocked a SunTag with 24 copies of GCN4 epitope into the amino terminus of the endogenous *nanos* CDS, referred to as *suntag-nanos* (Extended Data Fig. 1a, Methods and Supplementary Notes). We utilized a newly developed monomeric

msGFP2-fused scFv to detect the fully synthesized SunTag protein. This monomeric form evades the aggregation observed when SunTag was detected by the original super-folder GFP-fused scFv[45] (Extended Data Fig. 1b,c). Embryos were collected and fixed from female flies carrying *suntag-nanos*, germline-expressing *scFv-GFP* and *Vasa-mApple* as a germ granule marker[46]. The mRNA of *suntag-nanos* was hybridized using smFISH probes against the *suntag* sequence, and the embryos were imaged with confocal microscopy. The mRNA of *suntag-nanos* distribution within the embryo was similar to that of the native *nanos* mRNA: present throughout the embryo while enriched in the germplasm (Fig. 1b). Notably, we observed a substantial amount of GFP foci in the germplasm, while GFP foci were scarce elsewhere (referred to as soma) (Fig. 1b,c). A zoomed-in view showed that most of the GFP foci were co-localized with smFISH foci, representing individual translation sites (Fig. 1c and Extended Data Fig. 1b,c). To directly demonstrate that these GFP foci are sites of translation, we treated embryos with puromycin, a translation inhibitor that disassembles polysomes, or harringtonine, which blocks initiation and allows polysome run-off. Both inhibitors abolished the GFP foci, validating that the GFP foci represented actively translating polysomes (Extended Data Fig. 1d). We used FISH-QUANT software to locate individual RNA foci and GFP foci and determine whether an RNA molecule co-localized with a GFP focus, thus being translated[47]. We detected ~30–50% *suntag-nanos* mRNA being translated in the germplasm, whereas in the soma, the percentage was lower than 2% on average (Fig. 1c,d). Detecting SunTag in stage 1 embryos using anti-GCN4 immunostaining instead of scFv–GFP provided comparable results (Extended Data Fig. 2). The SunTag immunostaining signal resembled the spatial pattern of Nanos protein immunostaining in a wild-type embryo (Extended Data Fig. 2c). At stage 5 when PGCs are fully cellularized, there was a strong reduction of bright scFv–GFP foci within PGCs, seemingly suggesting that translation was repressed (Extended Data Fig. 2d,e). Anti-GCN4 immunostaining, however, confirmed that *suntag-nanos* translation was active after PGC formation and the reduction of GFP foci was due to the limitation and depletion of scFv–GFP within PGCs[45] (Extended Data Fig. 2b,d,e). Thus, active *suntag-nanos* translation was maintained throughout PGC formation (Extended Data Fig. 2b), consistent with the accumulation of Nanos protein in PGCs seen in previous studies[30,48]. Together, *suntag-nanos* mRNA exhibited germplasm-localized translation, consistent with the translation pattern of native *nanos* mRNA inferred from Nanos protein localization.

Translation of *nanos* mRNA in the germplasm is dependent on the assembly of germ granules and can occur at the anterior pole of an embryo if germ granules are ectopically formed there[35,48]. In agreement, translation of *suntag-nanos* at the posterior pole was abolished when germ granule assembly was perturbed by knocking down maternal *oskar* expression (Fig. 1e and Extended Data Fig. 3a). We induced germ granule assembly at the embryo's anterior pole by expressing transgenic *osk-bcd* 3′ UTR[48]. The mRNA of *suntag-nanos* localized to the anterior pole similar to the native *nanos* and was translated at a level comparable to native germplasm at the posterior (Fig. 1f,g and Extended Data Fig. 3b), validating the necessity and sufficiency of germ granules in activating *nanos* translation.

*Nanos* mRNA is synthesized during oogenesis and becomes localized to the germplasm in developing oocytes[30]. To further validate the translational regulation on *suntag-nanos*, we assessed its translation during oogenesis (Fig. 2a). Imaging translation in the germplasm of mature oocytes showed a substantially lower translation rate of *suntag-nanos* than in the embryonic germplasm (Fig. 2b,c), which is expected owing to the widespread translational dormancy of mature oocytes followed by translational activation after egg activation[30,49,50]. Furthermore, recapitulating egg activation in vitro by immersing mature oocytes in a hypotonic buffer was sufficient to activate translation in the germplasm[50] (Fig. 2d,e and Supplementary Video 1). By contrast, *suntag-nanos* translation in nurse cells is constitutively active (Fig. 2b,c), consistent with the abundance of Nanos protein in nurse cells observed previously[30,51]. Thus, *suntag-nanos* directly demonstrates the spatiotemporal pattern of *nanos* translation in vivo, which previously had only been deduced from the pattern of Nanos protein distribution and functional studies. This establishes *suntag-nanos* as a reliable tool to study translational control in germ granules.

## A 5′–3′ orientation of translating mRNA on germ granules

Using *suntag-nanos* as a single-molecule visual translation reporter in vivo, we investigated the spatial distribution of *nanos* mRNA translation relative to germ granules. By live imaging, we observed the co-movement of germ granules and their associated translation foci for over 6 min (Fig. 3a,b and Supplementary Video 2), suggesting a stable physical association between the polysomes and germ granules. To map the distribution quantitatively, we used smFISH images and established an image analysis pipeline to measure the distance between individual mRNA or GFP foci and their closest granule border. We first defined the border of individual germ granules by segmenting the Vasa signal with Ilastik, a machine-learning-based image analysis program[52] (Extended Data Fig. 4a). The coordinates of individual mRNA and GFP foci were determined and extracted using FISH-Quant, and their relative distance to the closest granule border was mapped (Extended Data Fig. 4b and Methods). Consistent with the fact that *nanos* mRNA localizes to germ granules and that low-abundance mRNAs tend to reside at the border of granules[53,54], we found that *suntag-nanos* mRNA (detected by *suntag* smFISH) was enriched around the granule border. As controls, we performed the same analysis on simulated random spots, randomized *suntag-nanos* mRNA foci by rotating the image of the smFISH channel, and smFISH foci of *osk* mRNA, which do not localize to germ granules[53,55]. Foci in all three control images exhibited a similar distribution that was not enriched around the border (Extended Data Fig. 4c–g). These results demonstrate that the distribution of *suntag-nanos* mRNA is non-random and centred around the granule border, confirming previous single-molecule studies[53,54,56].

---

**Fig. 3 | Spatial distribution of the polysome and orientation of translating mRNA. a**, Left: schematic of the live imaging setup. Right: a live image of germplasm. Blue, Vasa; green, scFv–GFP. **b**, Spatiotemporal tracking of germ granules and attached translation foci. Two pairs of germ granules (arrowheads) and translation foci (arrows) were tracked for 400 s, and showed co-movement throughout the movie. **c–f**, Orientation of translating mRNA. Translating *suntag-nanos* mRNAs in embryos from *Vasa-mApple/+; suntag-nanos, scFv-GFP/Df(nanos)* flies are detected with smFISH against *suntag* (**c**) and *nanos* 3′ UTR (**e**). Example germplasm images are shown in **c** and **e**. The orthogonal views of the outlined regions are shown on the right. Scale bar for the orthogonal views: 0.3 μm. Blue, Vasa; magenta, mRNA smFISH; green, scFv–GFP. The distributions of scFv–GFP and smFISH foci were mapped and plotted in relative frequency histograms overlaid with KDEs in **d** and **f**. The *x* axis refers to the distance of foci centroids to the border of the closest granule; zero marks the granule border; a negative value denotes being inside a granule and a positive value denotes outside. In total, 12,684 smFISH foci and 12,733 scFv–GFP foci from images of 7 embryos were mapped in **d**. A total of 5,663 smFISH foci and 5,649 scFv–GFP foci from images of 3 embryos are mapped in **f**. **g**, Detecting the 5′ and 3′ sequence of native *nanos* mRNA in wild-type (WT) embryos. The schematic shows the probes used for smFISH. 5′ probe signal (green) coats around the 3′ probe signal (magenta) and germ granule marker Oskar–GFP (blue). The orthogonal views of the outlined region are shown on the right. Scale bar, 0.3 μm. **h**, Averaging germ granule images showed the distribution pattern of 5′ and 3′ of *nanos* mRNA relative to germ granules. Using the Oskar channel as a reference, 40 images of germ granules from 3 embryos were randomly picked, made into a z-stack and averaged. **i**, PCC measurement showing a stronger co-localization of Oskar with *nanos* 3′ signal than 5′. *n* = 4 embryos for both conditions. The data are the mean ± s.d. Statistics: two-tailed Welch's *t*-test.

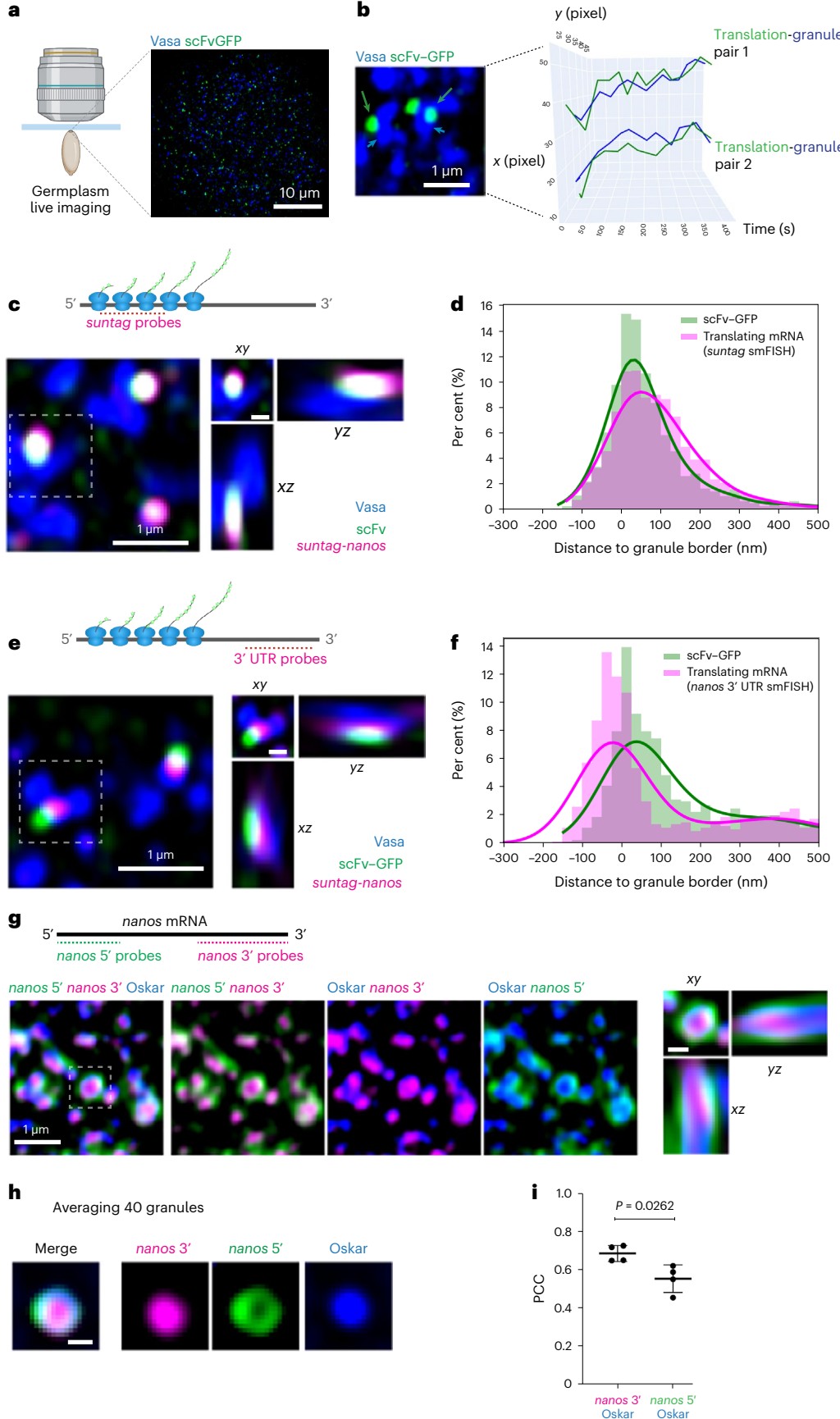

Next, we analysed the distribution of GFP foci to infer the position of polysomes relative to the granule border. GFP foci (that is, polysomes) were enriched around the granule border, which overlapped with the distribution of the *suntag* smFISH spots, consistent with their close physical association (Fig. 3c,d). This result demonstrates that *nanos* translation occurs close to the border of germ granules. However, when *suntag-nanos* mRNA was detected with smFISH probes targeting the *nanos* 3′ UTR, we noticed a shift between the smFISH and GFP distributions, reflecting the separation between the N-terminal coding sequence (CDS) of mRNA and 3′ UTR (Fig. 3e). Interestingly, relative to the GFP spots, *nanos* 3′ UTR smFISH spots were skewed towards the inside of the granule (Fig. 3f), suggesting that the 3′ UTR of translating *suntag-nanos* mRNA is preferentially buried inside germ granules. This in vivo observation is consistent with the *nanos* 3′ UTR being necessary and sufficient for mRNA localization to germ granules and the interaction between the *nanos* 3′ UTR and RBPs (Oskar and Aubergine) in germ granules[28,37,39,57–59]. Together, our analysis revealed a specific conformation adopted by germ granule-localized, translating *nanos* mRNA: the CDS and associated polysome of translating mRNA are oriented towards or exposed on the surface of granules, while the 3′ UTR is anchored internally.

As the outward positioning of the CDS may be caused by the *suntag* sequence, we asked whether native *nanos* mRNA adopted such an orientation. We used smFISH probes against the 5′ UTR and first 400 nt of CDS (5′ probe) and a 3′ UTR probe to detect the two ends of *nanos* mRNA simultaneously. Notably, while the 3′ UTR signal appeared as foci co-localized with germ granules, the 5′ probe signal formed a ring-shaped coating around the 3′ UTR foci and germ granules (Fig. 3g,h). We used Pearson correlation coefficient (PCC) to quantify the level of co-localization between the two fluorescent signals[53]. The 5′ probe signal showed significantly less co-localization with germ granules than the 3′ UTR probe (Fig. 3i). This observation suggests an outward orientation of the *nanos* 5′ end and internal anchorage of the 3′ UTR, consistent with the analysis of *suntag-nanos* data.

## mRNA positioning correlates with translation on germ granules

Next, we asked how the translational status of mRNA affected its distribution in germ granules by comparing the distributions of translating versus non-translating *suntag* smFISH foci (Fig. 4a). We noticed that *suntag* smFISH foci of non-translating mRNA distribute more towards the inside of granules, a pattern similar to that of the distribution of the 3′ UTR of translating mRNA (Fig. 4b,c), suggesting that the 5′ UTR and CDS of mRNAs appeared to reside inside granules when the mRNA was not being translated (Fig. 4d). By contrast, translating mRNAs tended to have the CDS localized to the surface of germ granules (Fig. 4b,c). Consistent with this model, in oocyte germplasm where *suntag-nanos* translation was largely repressed, *suntag* smFISH foci distribution showed a global inward shift towards the inside of the

granules compared with embryonic germ granules (Figs. 2c and 4e). The 5′ probe smFISH signal of native *nanos* mRNA also showed significantly stronger co-localization with the 3′ UTR and germ granules in oocyte compared to embryos (Fig. 4f–h). This suggests that the 5′ UTR and CDS of *nanos* mRNA are positioned inside germ granules in the untranslated state (Fig. 4d). We asked whether this distribution of translation was due to an exclusion of ribosomes from the interior of germ granules. We immunostained for a ribosomal protein RPS6 as a proxy for ribosomes. Within germplasm, RPS6 staining exhibited an enrichment both within and around germ granules (Fig. 4i,j), suggesting against the exclusion of ribosomes.

To test the causality between translation and exposure of the CDS, we used harringtonine to block translation at initiation and let polysomes run off. The total *suntag* smFISH signal distribution did not show a strong inward shift after harringtonine treatment (Fig. 4k,l). This suggests that translation may bring *nanos* 5′ outward during oocyte-to-embryo transition but is not required to maintain the outward orientation afterward. Alternatively, a translation-independent mechanism may drive and sustain the orientation of the 5′ end towards the granule margin during granule-dependent translation.

## Germ granules de-repress translation

After establishing that germ granules were the sites of *nanos* translation, we investigated the mechanism of translational activation in germ granules. The translational repression of *nanos* in the soma is mediated by the translational repressor Smaug, which binds to the Smaug response element (SRE) in the *nanos* 3′ UTR[32,60,61]. To explore Smaug-mediated translational regulation, we generated transgenic flies with *UAS*-driven *suntag-nanos* constructs that varied in their 3′ UTRs. Driven by a maternal Gal4 activator, the respective RNAs either carried a wild-type *nanos* 3′ UTR (*suntag-nanos-WT*), which showed germplasm-restricted translation, the same pattern as the CRISPR-generated *suntag-nanos*; a 3′ UTR with a mutated SRE (*suntag-nanos-SREmut*) that directed RNA localization to the germplasm but lacked binding sites for the Smaug repressor and exhibited significantly elevated translation in the soma; and a *tubulin* 3′ UTR that was evenly distributed throughout the embryo, did not bind Smaug and supported constitutive translation in embryos (*suntag-nanos-tub 3′* UTR) (Fig. 5a,b and Extended Data Fig. 5). Thus, these *suntag-nanos* constructs recapitulated the requirement of the *nanos* 3′ UTR for RNA enrichment in granules and the role of the SRE sequence for translational repression in the soma observed previously[32,60,61].

We used these constructs to analyse translational dynamics quantitatively at the single-molecule level. This revealed that, in germ granules, a similar fraction of *suntag-nanos-SREmut* mRNA was translated compared with *suntag-nanos-WT* (Fig. 5b). The fact that local translation was not significantly increased in the Smaug-binding site mutant indicates that SRE sequences do not mediate repressor activity in the germplasm[31,32]. However, the fraction of translating

**Fig. 4 | mRNA positioning correlates with translation on germ granules.**
**a**, Example image of translating (arrows) and non-translating (arrowheads) mRNA. Blue, Vasa; magenta, *suntag* mRNA; green, scFv–GFP. **b**, Relative frequency histogram with KDE curve of translating and non-translating mRNA distribution in germplasm. A total of 12,684 translating and 19,712 non-translating foci from 7 images over 7 embryos were plotted. **c**, The translating fraction in each bin of the *x* axis from seven embryos was calculated and plotted. The data are the mean ± s.d. The average translating fraction in the entire germplasm is indicated as the dashed line. The translating fractions on the granule surface ($0 \le x \le 200$ nm) were compared with the ones within granules ($x < 0$) or ones not localized to granules ($x > 400$ nm) using two-tailed Welch's *t*-test. **d**, A model of the predicted orientation and distribution of translating and non-translating mRNAs in germ granules. **e**, Distribution of total *suntag-nanos* mRNA stained by *suntag* probes in stage 1 embryos and stage 14 oocytes. A total of 6,468 foci from images of 4 oocytes and 32473 foci from images of 7 embryos

were mapped. **f**, Detecting the 5′ and 3′ sequence of native *nanos* mRNA in wild-type oocytes. The orthogonal views of the outlined region are shown on the right. Scale bar, 0.3 µm. **g**, PCC analysis between *nanos* 3′ and 5′ signals in oocytes and embryos. $n = 4$ embryos and 4 oocytes. **h**, PCC analysis of Oskar signal with *nanos* 5′ or 3′ signal in oocytes. Statistics in **g** and **h**: two-tailed Welch's *t*-test. $n = 4$ oocytes for both conditions. In **g** and **h**, the data are the mean ± s.d. **i**, Distribution of RPS6 (anti-RPS6, magenta) in germplasm. The germ granules are marked by VasaGFP (green)[46]. **j**, *z*-stacks of 40 images of germplasm with germ granules at the centre or without germ granules were made and *z*-projected by summing slices. Scale bar, 0.25 µm. **k**, *suntag* mRNA (magenta) distribution on germ granules (Vasa) in embryos treated with DMSO (control) and harringtonine. **l**, Relative frequency histogram with KDE of total *suntag* smFISH foci distribution in embryos treated with DMSO (2,618 foci from 5 embryos) or harringtonine (8,545 foci from 4 embryos).

*suntag-nanos-SREmut* mRNA was higher in the germplasm than in the soma (Fig. 5b). One possibility is that, in addition to Smaug, *nanos* translation is suppressed by other translational repressors[62], which are not mediated by SRE sequences in the 3′ UTR but also counteracted by germ granules. Alternatively, and not mutually exclusive, germ granules may also actively promote translation in addition to derepression. In line with this hypothesis, we observed an enrichment of eIF4G and PABP with germ granules (Fig. 5c,d). Moreover, it has been shown

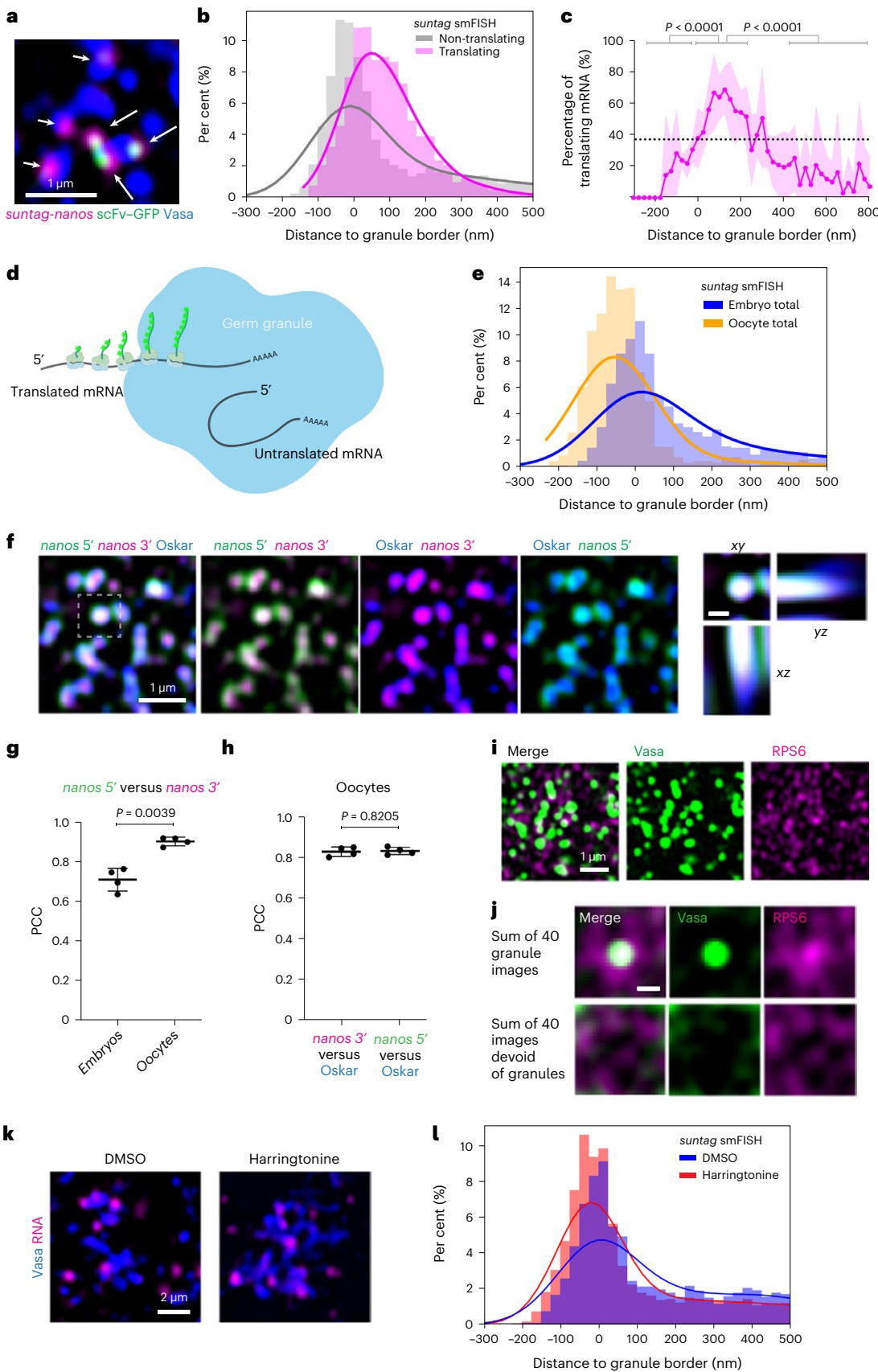

that eIF4A and eIF3 are recruited by the germ granule components Tudor and Aubergine[21,63]. Thus, in addition to a dominant derepression mechanism needed to overcome Smaug, select translation factors recruited by germ granules may facilitate mRNA translation. Together these results demonstrate a primary role for germ granules in protecting localized mRNA from translational repression, thereby allowing translation to occur.

Our results show that localized *nanos* mRNA was specifically translated on germ granules and that this was achieved primarily in germ granules by preventing translational repression. Next, we asked whether germ granules specifically modulate the kinetics of translation[40,64]. For example, germ granules may boost *nanos* translation by increasing translational initiation or elongation rate, apart from protecting *nanos* from translational repression by Smaug. We utilized the *suntag-nanos-SREmut* transgene to directly compare the translation kinetics of unlocalized mRNA in the soma with that of localized mRNA in granules within the same embryo. We measured the intensity of individual polysomes (SunTag staining) and found that the average intensities did not differ significantly between the germplasm and the soma (Fig. 5e). As polysome intensity is determined by the number of ribosomes loaded onto *suntag-nanos* mRNA[64,65], this result indicated that translating *suntag-nanos* mRNA have similar ribosome occupancy between germplasm and soma. We utilized the intensity of fully synthesized SunTag-Nanos peptide (dim SunTag foci without co-localized mRNA) to estimate the number of ribosomes on a translating mRNA[40] (Extended Data Fig. 6a,b and Methods). Translating *suntag-nanos* mRNA carried approximately one ribosome per 300 nucleotides in the CDS (Extended Data Fig. 6c), which is comparable to reported measurements carried out in tissue culture systems and *Drosophila* embryos[40–42,65,66].

To measure the elongation rate of translation in the soma and germplasm, we utilized fluorescence recovery after photo-bleaching (FRAP) of translation foci in live embryos[64] (Fig. 5f–h). We tracked individual translation foci in live embryos for over 5 min (Fig. 5f and Supplementary Video 3). Most tracked foci maintained their intensity over the live imaging process, indicating a steady state with constant translational initiation and elongation. After photo-bleaching individual foci, GFP fluorescence recovered over time and plateaued at the initial intensity (Fig. 5f,h and Supplementary Video 4). It has been reported that the binding of scFv–GFP to SunTag epitopes is stable, with a binding half-life of 5–10 min[67]. The full recovery of translation foci took around 4 min, indicating that the synthesis of new SunTag peptides, instead of the exchange of scFv–GFP, led to the fluorescence recovery (Fig. 5f). Comparing *suntag-nanos-SREmut* translation in the soma and the germplasm, we found that the FRAP curves of GFP spots closely matched (Fig. 5g and Supplementary Videos 4 and 5), suggesting similar elongation rates. We used a mathematical model to fit the FRAP data and calculate the elongation rate[41,44,64] (Extended Data

Fig. 6d,e and Methods), which yielded 3.71–4.28 amino acids per second in the germplasm and 4.19 amino acids per second in the soma respectively (Fig. 5f,g). These values are similar to the eukaryotic translation elongation rates calculated from ribosome profiling experiments and SunTag imaging in tissue culture[40,41,68,69]. Similar elongation rates and ribosome occupancy also suggest a similar steady-state translation initiation rate between germplasm and soma. Together, these results suggest that germ granules do not increase the steady-state initiation and elongation rates of translation. Instead, our results are consistent with the conclusion that germ granules allow the translation of *nanos* mRNA mainly by counteracting translational repression by Smaug.

## Oskar regulates the localization of translational repressors

It has been unclear how germ granules protect *nanos* mRNA from the repression by Smaug, which binds to the SRE sequences within the *nanos* 3′ UTR[32,61]. As we observed that the 3′ UTR of translating *suntag-nanos* mRNA was embedded inside germ granules, germ granules may create a space that excludes Smaug and consequently protects *nanos* 3′ UTR from Smaug binding and repression (Fig. 4d). By imaging transgenic Smaug–GFP embryos, we observed that Smaug was present throughout the embryos, forming heterogeneous clusters in the soma (Extended Data Fig. 7a). In the germplasm and PGCs, however, we unexpectedly found that Smaug was enriched within the germ granules, refuting the exclusion model (Fig. 6d,e and Extended Data Fig. 7a). It has been established that Smaug represses translation by recruiting the eIF4E-binding protein Cup and the CCR4-NOT deadenylation complex to inhibit translational initiation and assembling a stable repressive RNP complex with the P-bodies protein ME31B (DDX6 homologue)[34,36,62,70]. We examined the distribution of these Smaug co-factors (Cup, CCR4, NOT3 and ME31B) in germplasm and found none of them enriched within germ granules similar to Smaug (Extended Data Figs. 7b and 8), indicating that the germ granule-localized Smaug appears unable to recruit the necessary downstream effectors needed for translational repression. Specifically, we found that ME31B was localized on the surface of germ granules only after PGC formation, while in the soma ME31B forms distinct micrometre-size granules (Extended Data Fig. 7b). Thus, germ granule-localized Smaug may not be conducting its role as a translational repressor.

We reasoned that the selective localization of Smaug to germ granules should be controlled by particular granule protein components and might underlie the derepression of *nanos* mRNA. Smaug has been shown to interact with Oskar, the scaffold protein that drives the assembly of germ granules and recruits mRNA[35,38,39,71]. Oskar has an N-terminal LOTUS domain that mediates dimerization and binds to Vasa, a C-terminal SGNH-like domain with RNA-binding function, and a 159-residue-long linker region in between, which is predicted to be mainly intrinsically disordered[37,39] (Fig. 6a). Most of the *oskar*

**Fig. 5 | Kinetics of *suntag-nanos* translation. a**, Example image of the posterior of an embryo expressing Vasa–mApple (blue), *nanos-suntag-SREmut* (*suntag* smFISH, magenta). SunTag is detected by anti-GCN4 (green). An image of the anti-GCN4 channel is shown on the right with germplasm and soma outlined. **b**, Translating fractions in embryos from flies expressing transgenic *suntag-nanos-WT* (soma $n = 6$, germplasm $n = 7$), *suntag-nanos-SREmut* (soma and germplasm $n = 7$) or *suntag-nanos-tub* 3′ UTR (soma $n = 11$, germplasm $n = 7$). The data are the mean ± s.d. Pairwise statistical comparisons were conducted using two-tailed Welch's *t*-test. **c**, Left: the posterior of an embryo expressing Vasa–mCherry (magenta) and eIF4G–GFP (green), and zoomed images of germplasm (right), showing the enrichment of eIF4G to germ granules. **d**, Left: the posterior of an embryo expressing Vasa–mCherry (magenta) and yellow fluorescent protein (YFP)-tagged PABP (green), and zoomed images of germplasm (right), showing the association of PABP puncta with germ granules. **e**, The intensities of polysomes (anti-GCN4 staining) in soma and germplasm of embryos from flies expressing *UAS-suntag-nanos-SREmut*. Quantification results from four

embryos were plotted in a super-plot. Individual dots represent the intensities of individual polysomes, each colour-coded by the embryo. Each coloured circle represents the mean intensity of each embryo. The black lines and error bars are the mean ± s.d. of the four embryos. Statistical comparison was performed on the mean intensities of individual embryos using two-tailed Welch's *t*-test. **f**, The intensities of polysomes over time during live imaging *suntag-nanos* mRNA translation with (red curve, mean ± s.d. of 10 curves from 3 embryos) or without (blue curve, mean ± s.d. of 23 curves from 5 embryos) photo-bleaching when time is 30 s. The elongation rate calculated from the plot is indicated. **g**, Polysome intensities of *suntag-nanos-SREmut* mRNA in germplasm (blue curve, mean ± s.d. of 35 curves from 5 embryos) and soma (red curve, mean ± s.d. of 19 curves from 6 embryos) over time with photo-bleaching when time is 30 s. The elongation rates calculated from the plot are indicated. Note that elongation rates are not notably different between wild-type and *SREmut* RNA. **h**, Representative time-lapse image of FRAP of two translation sites (arrowheads). Blue, Vasa; green, scFv–GFP. Scale bar, 500 nm.

loss-of-function alleles identified so far have mutations within the LOTUS and SGNH-like domain and show defects in germ granule formation and RNA localization, precluding the analysis of Oskar's potential function as a translational regulator[22,72]. The linker sequences of different *Drosophila* species are not conserved but enriched in the amino acids asparagine (Asn, N) and glutamine (Gln, Q) (Fig. 6b and Extended Data Fig. 9a), which are over-represented in prion-like proteins[73,74]. To probe the functional importance of these sequence features of Oskar in Smaug localization and *nanos* translation, we

created a mutant Oskar protein where all the asparagine and glutamine residues in the linker region were mutated to glycine (Oskar-NQmut). We expressed the mutant protein at the anterior of the embryos via a *UAS-Oskar-NQmut-bcd 3'* UTR transgene so that native germ granules at the posterior can serve as an internal wild-type control (Fig. 6c). *Oskar-NQmut-bcd 3'* UTR formed germ granules with comparable morphology and localized *nanos* mRNA similar to *Oskar-WT-bcd 3'* UTR (Extended Data Fig. 9b). Furthermore, Oskar-NQmut granules exhibited similar physical properties as wild-type germ granules on

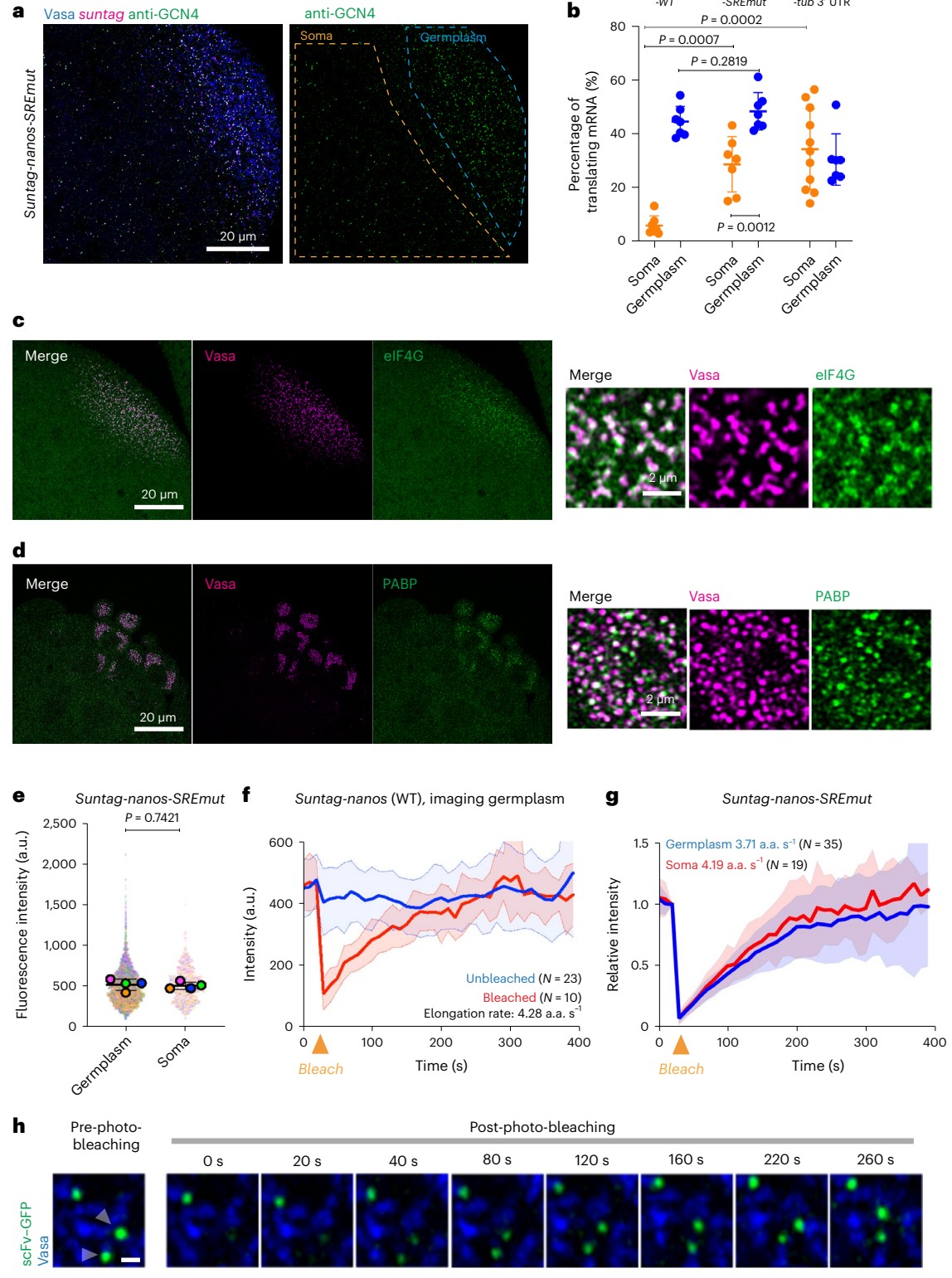

the basis of FRAP assay (Extended Data Fig. 9c). Thus, the Asn/Gln residues in the Oskar linker region were not essential for mediating germ granule assembly, modulating material properties or recruiting *nanos* mRNA. However, Oskar-NQmut germ granules completely lost their enrichment for Smaug (Fig. 6d,e), suggesting that the Oskar linker region mediates the recruitment of Smaug to germ granules, which is disrupted by Asn/Gln -to-Gly mutations in the sequence.

Next, we investigated whether Oskar-NQmut affects *nanos* translation. To this end, we quantified the translation of *suntag-nanos* mRNA localized to the anterior Oskar-WT and Oskar-NQmut germ granules. We found a roughly 50% decrease in the percentage of translating *suntag-nanos* mRNA in germ granules composed of Oskar-NQmut protein compared with wild-type granules, suggesting a compromised translational function caused by this mutant (Fig. 6f,g). Consistent with reduced translation, the segmentation phenotypes caused by anteriorly expressed Nanos protein were much milder in *Oskar-NQmut-bcd* 3′ UTR embryos than in *OskarWT-bcd* 3′ UTR embryos, validating that less Nanos protein was produced by Oskar-NQmut granules[29,35,48] (Fig. 6h). The reduction in translation could be due to a direct failure of Oskar protein to activate translation or to a loss of the ability to counteract Smaug-meditated repression in germ granules. We found that *suntag-nanos-SREmut* mRNA, which is not subject to repression by Smaug, was translated in Oskar-NQmut germ granules at a similar level as in wild-type germ granules (Fig. 6g), supporting the hypothesis that Oskar-NQmut granules are specifically compromised in their ability to counteract the repression by Smaug. Together, these results suggest that Oskar controls *nanos* translation by mediating the selective sequestration of Smaug in germ granules (Fig. 6i).

## Discussion

It has been unclear whether biomolecular condensates can activate translation by directly serving as compartments for translation. Here, we utilized the single-molecule imaging method, SunTag, to visualize the translation of *nanos* mRNA in vivo to demonstrate that *Drosophila* germ granules are the sites for active translation while unlocalized mRNA is subject to translational repression. The SunTag system and high-resolution microscopy revealed the conformations adopted by translated and untranslated mRNA on germ granules. The quantitative nature of the SunTag system allowed us to dissect how germ granules affect translation efficiency and steady-state kinetics in vivo, which is not possible to unravel by conventional biochemical approaches. By mutating the disordered linker region of the scaffold protein Oskar, we uncovered its role in controlling the selective sequestration of translational repressors in germ granules and, thereby, permitting *nanos* mRNA translation.

Our data distinguish *Drosophila* germ granules from most of the well-studied RNP granules that store translationally repressed mRNA and have to be disassembled to resume mRNA translation. However, *Drosophila* germ granules might not represent a unique case of RNP granules providing space or a platform for translation[4]. In fermenting yeast cells, mRNAs encoding glycolytic enzymes co-localize in specialized RNP granules and probably undergo translation within these granules[19]. In the PGCs of zebrafish embryos, *nanos3* mRNA is suggested to be translated at the periphery of germ granules based on the distribution of ribosomes[75]. *Pou5f3* mRNA granules in zebrafish embryos also have been shown to co-localize with nascent Pou5f3 peptides, suggesting the granules as translation sites[18]. In mouse spermatids, liquid–liquid phase separation of FXR1 is essential for translational activation of FXR1 target mRNAs, suggesting that FXR1-RNA condensates are the compartments for activated translation[17]. Interestingly, many of these examples were found in adult germ cells or early embryos, where transcription is largely inactive and translational regulation dictates the temporal and spatial distribution of proteins. Numerous specialized RNP granules or phase-separated condensates in germ cells and early embryos have been described so far[76]. Thus, we expect more cases of translationally active RNP granules to be uncovered, establishing translational activation by RNP granules as a prevalent mechanism regulating gene expression.

High-resolution imaging allowed us to locate translating polysomes around the border or the surface of germ granules with 3′ UTRs embedded internally. This extroverted orientation of polysomes is unlikely to be due to the lack of accessibility for translation machinery within RNP granules because we detected ribosomes and initiation factors inside granules. However, we propose that translation initiation, which requires sophisticated collaboration among multiple protein complexes, may be unfavoured within a highly condensed environment[77,78]. The correlation between translation status and the location of mRNA CDS suggests a potential regulatory mechanism: by controlling the inward or outward movement of a CDS, translation can be tuned up or down. Our results also suggest that translating mRNA adopts an extended conformation, while untranslated mRNA becomes folded or compacted. Similar observations have been made in cell culture by looking at the 5′ and 3′ ends of various mRNAs[79–81]. These results suggest that the classic closed-loop model of mRNA may not represent the predominant and stable conformation of translating mRNA[82].

Notably, only 4% of total *nanos* mRNA localize to germ granules, while the remaining 96% are spread throughout the embryo's soma[31,53]. Inappropriate translation of *nanos* mRNA causes embryonic polarity and segment patterning defects[29]. Therefore, strict translational repression of unlocalized mRNA and effective derepression by germ granules is necessary to establish the Nanos morphogen gradient emanating from the embryo posterior[29,31,32,36,60,61,70]. Our imaging with *suntag-nanos* unambiguously demonstrates the repression–derepression dichotomy between germplasm and soma. Furthermore, we

**Fig. 6 | Oskar linker region controls Smaug localization and *nanos* translation. a**, AlphaFold structure model of short Oskar protein, with LOTUS domain in red, SGNH-like domain in blue and linker region in green. **b**, Percentage of glutamine (Q) and asparagine (N) in three regions of short Oskar proteins from 11 *Drosophila* species. Each dot represents the Oskar protein of a particular *Drosophila* species. **c**, Embryos expressing *Oskar-WT/ NQmut-bcd* 3′ UTR are immunostained with anti-Oskar antibody. **d**, Distribution of Smaug in germplasm. Images of germplasm induced by ectopic Oskar-WT or Oskar-NQmut at the anterior pole. Germ granules are labelled by Vasa–mApple (magenta). Smaug is visualized with Smaug–GFP (green). **e**, Intensity profiles of Vasa–mApple (magenta) and Smaug–GFP (green) along the lines across the germ granules induced by Oskar-WT or Oskar-NQmut. The data are the mean ± s.d. of 20 germ granules from 3 embryos for each genotype. **f**, Representative images showing the translation of *suntag-nanos* mRNA in germplasm induced by Oskar-WT or Oskar-NQmut. Blue, Vasa; magenta, suntag smFISH; green, anti-GCN4. **g**, Fraction of *suntag-nanos-WT* or *suntag-nanos-SREmut* mRNA translated in anterior germplasm induced by Oskar-WT or Oskar-NQmut. Each dot represents the normalized measurement of an embryo where the translating fraction in the anterior germplasm is divided by the translating fraction in the native germplasm at the posterior. Statistical comparisons between Oskar-WT and NQmut were performed by two-tailed *t*-test. $n_{nanosWT-OskarWT}$ = 7, $n_{nanosWT-OskarNQ}$ = 6, $n_{nanosSRE-OskarWT}$ = 7, $n_{nanosSRE-OskarNQ}$ = 5. The data are the mean ± s.d. **h**, Cuticle phenotypes generated by *Oskar-WT/NQmut-bcd* 3′ UTR. The images show a range of cuticle phenotypes corresponding to different levels of anteriorly expressed Nanos protein. The bar graph shows the frequency of each cuticle phenotype caused by *Oskar-WT/NQmut-bcd* 3′ UTR expression. Statistical comparison was performed using chi-square test. **i**, Oskar mediates Smaug localization and translational derepression of *nanos* mRNA. With wild-type Oskar, Smaug, but not its co-factors for translational repression (Cup/CCR4-NOT), localized to germ granules. Localized Smaug is dysfunctional in translational repression, allowing the translation of *nanos* mRNA. In Oskar-NQmut germplasm, Smaug loses localization in germ granules but gains functionality inside germ granules, thus repressing the translation of *nanos* mRNA.

demonstrate that the activation is achieved through increasing the fraction of translating mRNAs instead of alterations in ribosome occupancy and elongation rate. Our results are consistent with previous studies in tissue culture suggesting that the fraction of translating mRNA is highly variable among different mRNAs and strongly affected by spatiotemporal regulation[64]. Thus, controlling the translating fraction of

an mRNA rather than tuning translation efficiency could be a common and critical aspect of translational regulation, including in biomolecular condensates.

Composition control has an essential role in regulating the functions of biomolecular condensates. RNP granules often comprise a complicated set of RBPs, largely associated with translational

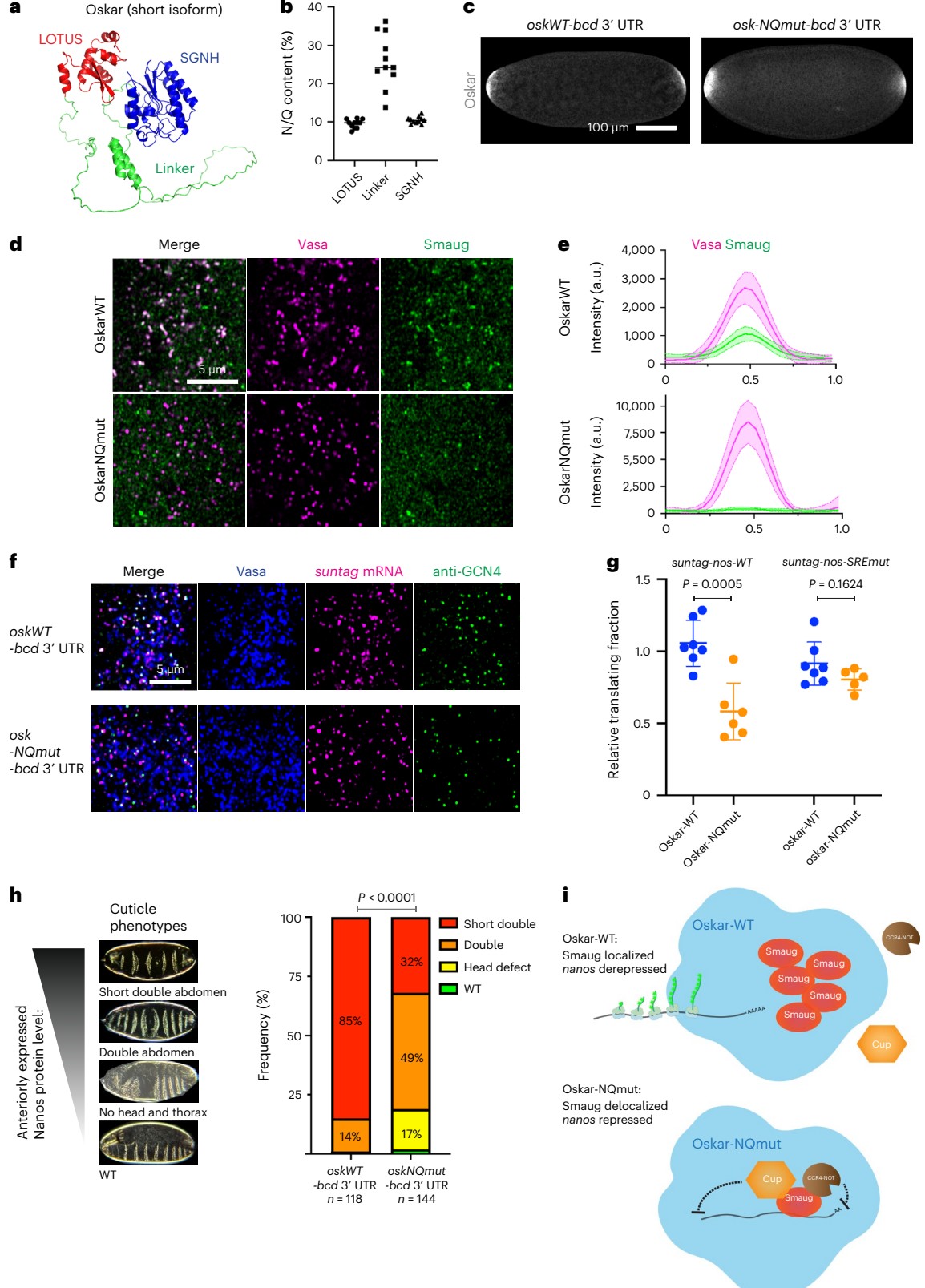

repression[83]. Consequently, RNP granules were considered translationally silent before rigorous tests using single-molecule imaging revealed some translation in stress granules[69]. Our work shows that the enrichment of translational repressors does not necessarily render RNP granules translationally repressive but can instead underlie the translational derepression mechanism of the target mRNA. Similarly, translation-activating condensates can form by RBPs that have long been considered as repressors such as FXR1[17,84]. Thus, the repression function of RBPs can be context dependent and might become inactivated or altered when localized or assembled into condensates. It remains unclear how germ granule-localized Smaug loses its repressor function and how Oskar mediates this effect. We speculate that, within germ granules, Smaug may lose interactions with co-factors such as ME31B, Cup or CCR4 or change to a conformation that disfavours RNA binding, potentially via its specific interaction with the Oskar linker region[71]. Alternatively, interactions with germ granule proteins can constrain the mobility of Smaug and, thus, limit its access to the target mRNA *nanos* (Fig. 6i). Testing these hypotheses requires bottom-up reconstitution with purified proteins in vitro and characterization of germ granule-specific RBP interactomes in vivo.

## Online content

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

## Methods

### Fly stocks

Fly stocks were maintained at 25 °C. Detailed experimental genotypes and sources of fly stocks are listed in Supplementary Table 1.

### Cloning, gene editing and transgenesis

All primers are listed in Supplementary Table 2. All the constructs were made using In-Fusion cloning (Takara Bio). All PCR was performed using CloneAmp HiFi PCR premix (Takara Bio 639298).

For CRISPR, the SunTag array was knocked-in to the *nanos* locus by homology-directed recombination following CRISPR–Cas9 gene targeting[85]. For generating the recombination template, the *nanos* sequence was PCR amplified from the genomic DNA of the *yw Drosophila* line. The SunTag sequence was PCR amplified from plasmid 5′TOP-SunTag-Renilla (J. Chao[86]; Addgene 119946). The plasmid backbone and DsRed selection marker were PCR amplified from pScarless-HD-DsRed (K. O'Connor-Giles, Addgene 64703) and assembled with *nanos* and SunTag fragments using In-Fusion assembly. SunTag array was placed right after the start codon of *nanos* open reading frame (ORF), while the DsRed marker was inserted after a TTAA sequence of the first intron of *nanos* to allow transposase-mediated excision. Two guide RNAs (guide 1: GATAACCGTAACTTTCGACC; guide 2: GTAAGAAGAAATGGCGAATA) were separately cloned into pCFD-dU6:3gRNA (DGRC_1362) by linearizing the plasmid with primers appended with the guide RNA sequences and ligation with KLD enzyme mix (NEB M0554S). The recombination template and two guide RNA plasmids were injected into various Cas9-expressing lines (BestGene). Transformant flies were screened using the DsRed eye marker. See Supplementary Information for details about the generation of the *suntag-nanos* CRISPR line.

**UAS-suntag-nanos transgenes.** To generate the *UAS-suntag-nanos* construct, the *nanos* sequence starting from the 5′ UTR to 500 nucleotides following the end of the 3′ UTR was PCR amplified from genomic DNA of *yw* strain and inserted into PCR-linearized pUASz1.1 plasmid (A. Spradling[87]; DGRC_1433), via In-Fusion assembly. The 5′ UTR of *nanos* is placed right after the Hs promoter of the plasmid so that the myosin IV intron (IVS) and Syn21 elements of the pUASz1.1 plasmid are removed. The P10 3′ UTR of the pUASz1.1 plasmid was also left out during cloning. The SunTag sequence was amplified from 5′TOP-SunTag-Renilla (Addgene 119946) and inserted after the *nanos* start codon via In-Fusion assembly.

To introduce mutations at the two SRE sequences in the *nanos* 3′ UTR[51,61], a gBlock of the sequence containing the mutant SREs was synthesized and replaced the wild-type sequence in the *UAS-suntag-nanos* plasmid. The wild-type sequences are SRE1: GCAGAGGCT<u>C</u>TGGCAGCTTTTGC and SRE2: AAATAGCGCC<u>C</u>TGGCGCGTTCGAT. In the mutant the underlined C is mutated to G, and the underlined G is mutated to C).

The sequence of tubulin84B 3′ UTR was amplified from genomic DNA and replaced the *nanos* 3′ UTR of *UAS-suntag-nanos* plasmid to generate *UAS-suntag-nanos-tub* 3′ UTR[27].

**UAS-osk-bcd 3′ UTR transgenes.** Full-length Oskar CDS and 3′ UTR of *bicoid (bcd* 3′ UTR*)* without the Nanos response element were PCR amplified from plasmid UAS-oskar-mCherry3xFLAGHA-bcd 3′ UTR[88]. pUASz1.1 vector was PCR linearized and assembled with *oskar* CDS and *bcd* 3′ UTR fragment to generate *UAS-osk-bcd* 3′ UTR. To generate *oskar-NQmut*, a gBlock of Oskar linker region containing N/Q-to-G mutations was synthesized and replaced the wild-type sequence of *UAS-osk-bcd* 3′ UTR plasmid.

For transgenesis, individual plasmids were injected into attP2 or attP40 lines (BestGene), and transformants were screened for the presence of the *mini-white* eye markers.

### Immunofluorescence

*Drosophila* embryos were collected for 3 h on an apple juice plate, dechorionated by incubating with 50% bleach solution for 2 min, extensively washed and transferred to a scintillation vial containing a 1:1 (v/v) mixture of heptane and 4% paraformaldehyde in PBS, in which embryos were permeabilized and fixed for 20 min. The paraformaldehyde was removed with a Pasteur pipette, followed by adding methanol and vigorous shaking for 15 s to remove the vitelline membrane. Embryos were washed three times for 5 min with methanol before being stored at 4 °C in methanol. Embryos were rehydrated by washing for 5 min with 50% methanol with 0.3% Triton X-100 in PBS (Triton/PBS) and then washed and permeabilized for 3× 15 min in 0.3% Triton/PBS Embryos were blocked with 1% bovine serum albumin (BSA) in 0.3% Triton/PBS for 30 min and subsequently incubated with primary antibodies diluted in the blocking solution (anti-Oskar (gift from P. Lasko) 1:1,000, rabbit anti-Nanos (Lehmann Lab) 1:1,000, rabbit anti-CCR4 and anti-NOT3 (gift from E. Wahle) 1:1,000, rabbit anti-RPS6 (Cell Signaling 2217) 1:200, rabbit anti-GCN4 (Novus Bio C11L34 1:1,000)) overnight at 4 °C. Embryos were washed five times for 10 min with 0.3% Triton/PBS, blocked for 30 min and incubated with secondary antibodies (Thermo Fisher Scientific anti-rabbit Alexa Fluor 488, anti-rabbit Alexa Fluor 647 and anti-rat Alexa Fluor 555) with 1:1,000 dilution for 4 h at room temperature. Then embryos were washed five times for 10 min with 0.3% Triton/PBS, stained with 4′,6-diamidino-2-phenylindole (DAPI) and mounted with ProLong Glass mounting medium (ThermoFisher, P36980).

### smFISH

The smFISH protocol with fixed embryos and ovaries is modified from that described previously[89,90]. Stellaris RNA FISH probes against *suntag*, *nanos* and *oskar* sequences were used for hybridization. The *nanos* CalFluor 590 and *oskar* CalFluor 590 probes have been described and used in previous work[53]. The *suntag* Quasar 670 and *nanos* 3′ UTR Quasar 670 probes were synthesized by LGC Biosearch Technologies. *Nanos* 5′ probe was produced through an enzymatic method using oligos purchased from IDT, Atto565-NHS (Atto-tec GmbH) and NH2-ddUTP (Lumiprobe)[91]. The probe sequences are listed in Supplementary Table 3. To perform smFISH on fixed embryos, stored embryos were rehydrated by washing for 5 min with 50% methanol with 0.1% Tween in PBS (Tween/PBS) and washing three times for 5 min in 0.1% Tween/PBS. Embryos were then washed with pre-hybridization buffer containing 2× SSC and 10% formamide (Fisher Scientific, AM9342) for 10 min at room temperature. The embryos were then incubated at 37 °C for 3 h in the hybridization mix (60 µl hybridization mix per sample with 50–100 embryos) containing 2× SSC, 10% (v/v) deionized formamide, 0.1 mg ml$^{-1}$ *Escherichia coli* transfer RNA, 0.1 mg ml$^{-1}$ salmon sperm DNA, 10 mM ribonucleoside vanadyl complex (NEB, S1402S), 2 mg ml$^{-1}$ BSA, 80 ng Stellaris probes and 10% (v/v) dextran sulfate. After hybridization, embryos were washed with pre-hybridization buffer twice for 15 min at 37 °C. The embryos were washed with 0.1% Tween/PBS three times for 5 min, stained with DAPI and mounted with ProLong Glass mounting medium.

When anti-GCN4 is used to detect SunTag protein, immunofluorescence was performed after smFISH. Following the 2×15 min washes with pre-hybridization buffer, embryos were washed and permeabilized with 0.3% Triton/PBS for 45 min. Embryos were blocked with 1% BSA in 0.3% Triton/PBS for 30 min and then incubated with rabbit anti-GCN4 (Novus Bio.) with 1:1,000 dilution overnight at 4 °C. Embryos were washed with 0.3% Triton/PBS five times for 10 min, blocked for 30 min and incubated with anti-rabbit secondary antibody (1:1,000) for 4 h at room temperature. Embryos were then washed with 0.3% Triton/PBS five times for 10 min, stained with DAPI and mounted with ProLong Glass mounting medium.

To detect the translation of *suntag-nanos* in ovaries, *Vasa-mApple/+; suntag-nanos, scFv-GFP/+* flies were used and ovaries were

hybridized with *suntag* smFISH probes. Ovaries were dissected in Robb's buffer (100 mM HEPES, 100 mM sucrose, 55 mM sodium acetate, 40 mM potassium acetate, 10 mM glucose, 1.2 mM magnesium chloride and 1 mM calcium chloride) and fixed with 4% paraformaldehyde in Robb's buffer for 20 min. After fixation, ovaries were washed with 0.3% Triton/PBS twice for 5 min, 50% methanol in 0.3% Triton/PBS for 5 min and 100% methanol for 30 min. Ovaries can be stored in methanol at 4 °C. The rehydration, hybridization and washing for smFISH follow the same protocol as for embryo samples described above.

The detailed genotypes of flies and reagents for fluorescence labelling (immunofluorescence/smFISH) used in each experiment are listed in Supplementary Table 1. Specifically, to detect *suntag-nanos* mRNA using probes against *nanos* 3′ UTR, embryos from *Vasa-mApple/+; suntag-nanos, scFv-GFP/Df(3R)DlSP* flies are used, where *Df(3R)DlSP* is a deficiency line for the *nanos* locus and, thus, the *suntag-nanos* allele is the only source of the *nanos* mRNA in embryos[30]. For experiments involving *UAS-suntag-nanos-SREmut, UASz-suntag-nanos-tub* 3′ UTR and *UAS-oskWT/NQmut-bcd* 3′ UTR, which increased *suntag-nanos* translation, we used anti-GCN4 to detect the SunTag instead of scFv–GFP owing to a potential depletion of scFv–GFP in the embryos[45].

## Confocal microscopy of fixed embryo samples

Images of whole embryos were acquired using Zeiss LSM780 confocal microscope with 10× 0.3 numerical aperture (NA) air objective and 2.2 pixels μm⁻¹. DAPI was excited by a 405 nm laser. Red fluorophores (mApple, mCherry, Alexa Fluor 555 or Alexa Fluor 561) were excited by a 561 nm laser. Green fluorophores (GFP, yYFP or Alexa Fluor 488) were excited by a 488 nm laser. The far-red fluorophore (Alexa Fluor 647) was excited by a 633 nm laser. Embryos were staged using DAPI staining[92].

High-resolution images of germplasm were acquired using Zeiss LSM980 confocal microscope with Plan-Apochromat 63×/1.4 NA oil objective with AiryScan 2 detector and SR mode. GFP and Alexa Fluor 488 were excited using a 488 nm laser; mApple, mCherry, Alexa Fluor 555 and Alexa Fluor 568 were excited using a 561 nm laser; Alexa Fluor 647 and Quasar 670 were excited using a 639 nm laser. Images were acquired with 1.7× zoom, 23.5 pixels μm⁻¹, 8 bits per pixel and without averaging (result imaging size 78.2 μm × 78.2 μm, 1,840 pixels × 1,840 pixels). Multiple *z*-stacks (~10–30 stacks) were taken with a 150 nm interval (voxel size 42.5 nm × 42.5 nm × 150 nm). Raw images were first processed with the 3D AiryScan processing function in ZEN Blue software. Imaging TetraSpeck fluorescent microspheres showed clear chromatic aberrations among three channels. Therefore, aberration correction files were generated using the channel alignment function in ZEN by correcting the signal misalignments in a microsphere image. The correction files were applied to correct the embryo images (post-Airyscan processing) using the channel alignment function. The images after Airyscan processing and channel alignment were saved as final data and used for later analysis and publication.

## Image analysis of translation foci and quantification

SunTag image analysis was performed using MATLAB-based software FISH-Quant_v3, which allows the detection of focal signals and analysis co-localization in a three-dimensional (3D) space[47]. Images taken from Zeiss LSM980 using the 63× oil lens and AiryScan 2 detector with three channels (germ granules marked by Vasa, *suntag* mRNA smFISH, SunTag protein stained by scFv–GFP or anti-GCN4) were first split using Fiji software. The Vasa channel images were used to define the outline of germplasm and soma. To detect the foci of *suntag* mRNA and SunTag protein (anti-GCN4/scFv–GFP), a pre-detection was performed to test a range of threshold values and determine the number of detected spots for each tested value. The number of detected spots usually plateau at a range of tested thresholds and the number increased exponentially with lower thresholds which indicated the detection of background or noise signals. The threshold was placed at the left side of the plateau range before the increase occurred and foci were detected with this set threshold. The detected spots were then fit with a 3D Gaussian, which determined the 3D coordinates and intensities of individual foci for later analysis. The co-localization between the detected mRNA and protein foci was analysed by the DualColor program of FISH-Quant. We set 400 nm as the maximum distance between two spots to be considered co-localized although the number of co-localization events usually plateaus at 250 nm. This analysis provides the percentage of mRNA foci co-localized with protein foci, which represents the percentage of translating mRNA.

## Distance measurement

To measure distances between the mRNA and SunTag foci and the germ granule surface, images taken from Zeiss LSM980 using 63× oil lens and AiryScan 2 detector with three channels (germ granules marked by Vasa–mApple or Vasa–GFP, *suntag-nanos* mRNA stained by smFISH probes against *suntag* or *nanos* 3′ UTR, SunTag protein stained by scFv–GFP or anti-GCN4) were used. Images were first analysed with FISH-Quant_v3 and DualColor as described in the section above, which provides a result file with the *x,y,z* coordinates of mRNA and SunTag foci and identifies translating and non-translating mRNA.

The machine learning-based image analysis program ilastik was used to segment germ granules visualized in the Vasa channel[52]. The classifier in the Pixel Classification workflow was trained using three representative images and was then applied to unseen images to perform binary segmentation of germ granules. The segmented germ granule files were then imported into FIJI and analysed using a FIJI macro. In brief, FIJI macro outputs files with the coordinates of each pixel categorized as belonging to germ granules based on the previous ilastik-based segmentation. Various pixel lists were compiled, which separated the pixels on the 3D surface of the granule (identified using the 3D Object Counter plugin in FIJI) from those entirely within the granule. Additionally, only the granules that were entirely within the acquired image (in all three dimensions) were used in the analysis. In other words, the granules that contacted the *x*, *y* or *z* border of the image were identified and excluded from the analysis because the borders of the image provide artificial surfaces for the granules and may affect the outcome of the analysis.

The 3D segmentation of germ granules and coordinates of SunTag and mRNA foci acquired from FISH-Quant analysis were then further analysed in a custom-built Python workflow to perform the 3D distance measurement of mRNA and SunTag foci from the surface of the germ granule. First, the minimum distance of each point to the closest pixel on the 3D surface of the granule was calculated. Next, any mRNA or scFv foci that were closest to a granule that touched the edge of the *z*-stack were excluded. This step was performed to ensure that any analysis on localization was only performed on foci associated with granules that were fully captured within the image. Then, the foci were categorized as being inside or outside the granule on the basis of their relative position to the 3D surface pixels of the granule compared with pixels entirely within (or outside) the granule. After the categorization of foci as either inside or outside granules, the minimum distance to the surface of the germ granule values was adjusted to be negative if the foci were inside the granule and kept as positive if the foci were outside the germ granule. The adjusted distance values were then plotted as a relative frequency histogram using Seaborn in Python. A bin size of 25 nm was used. Kernel density estimate (KDE) plots were generated and overlaid with the histogram.

## Computation controls of 3D distance measurements

A representative section of germplasm with relatively even coverage of germ granules across the image was first cropped from an image. The 3D distances of mRNA foci from the surface of the germ granules in this image were obtained and plotted as detailed in the 'Distance measurement' section. Then, the mRNA channel of the image was rotated 180° and analysed and plotted using the same workflow.

Next, 1 million points (located within the volume of the image) were generated by drawing from a uniform distribution in $x$, $y$ and $z$. These simulated points were then analysed using the same workflow previously mentioned.

### Distribution of 5' and 3' sequence of endogenous *nanos* mRNA and PCC quantification

We performed smFISH using *nanos* 5' Atto565 probe (targeting 5' UTR and the first 400 nt of CDS) and *nanos* 3' UTR Quasar 670 probe (targeting 3' UTR) on embryos or oocytes from Oskar–GFP CRISPR knock-in line (gift from G. Gonsalvez). After the smFISH protocol, the posterior ends of embryos or oocytes were cut off so that the posterior could face the coverslip after mounting to allow imaging of the germplasm centre. Germplasm centres, instead of the periphery, were used for the analysis specifically because they have a stronger Oskar protein signal but fewer RNA in each *nanos* homotypic cluster, allowing a better resolution of 5' signals. We still cannot detect separated 5' signals from individual mRNAs in a germ granule because of the limitation in the resolution of the microscope and the clustering of the mRNA. Nevertheless, the resolution was good enough to see the separation of 5' signals from 3' signals and Oskar–GFP signals. PCC measures the degree of co-localization of two objects with distinct fluorescence[93]. Images underwent background subtraction using Fiji. PCC was then measured using the BIOP-JACoP plug-in of Fiji program[94]. Thresholding was automatic, and the thresholding methods do not affect PCC outcomes.

### Puromycin injection

Embryos from *Vasa-mApple*/+; *suntag-nanos*, *scFv-GFP*/+ flies were dechorionated using 50% bleach for 2 min and washed thoroughly with water. About 40–50 embryos were then lined up and mounted at the edge of a coverslip by heptane glue with their posterior poles pointing towards the edge of the coverslip. Mounted embryos were placed in a desiccator at 18 °C for 10 min and then covered by halocarbon 700 oil. Then, 20 mM HEPES (control) or 10 mg ml⁻¹ puromycin in 20 mM HEPES (Gibco, A1113803) was injected at the posterior pole of the embryos using FemtoJet (Eppendorf, 5252000021) with Femtotips II (Eppendorf, 930000043) needles at 18 °C. The exact injected volume of solution was difficult to control, but generally, the volume was small to avoid cytoplasm to leak from the embryos. The injected embryos were aged for 15–30 min at 18 °C before being transferred to a glass vial containing a 1:1 (v:v) mixture of heptane and fixative (4% paraformaldehyde in PBS) and fixed for 60 min at room temperature. After fixation, embryos were transferred onto a double-sided tape within a Petri dish and covered with 0.1% Tween/PBS. The vitelline membrane was removed manually with the needle of an insulin syringe. Devitellinized embryos were stepped into 100% methanol and washed in methanol three times for 5 min before being stored or proceeding to smFISH.

### Harringtonine treatment

Harringtonine treatment does not require injection. Dechorionated embryos were incubated in a scintillation vial with heptane and 20 mM HEPES pH 7.4 solution containing either dimethyl sulfoxide (DMSO) or 100 µg ml⁻¹ harringtonine for 30 min. Heptane permeabilized the embryos to allow harringtonine to penetrate the embryos. Treated embryos were transferred into a vial with Heptane and fixative to undergo normal fixation protocol.

### Live imaging and FRAP

We found that fertilized embryos showed apparent cytoplasmic movement during live imaging, potentially due to the mechanical force generated during nuclear division and migration[95]. We found that unfertilized eggs had much less cytoplasmic movement, which allowed tracking individual polysomes for extended periods (>5 min), and, thus, were used for the live imaging.

For live imaging of *suntag-nanos* mRNA translation, we expressed *UAS-suntag-nos* (WT or SREmut) with a weak Matα-GAL4 (without VP16) maternal driver line to prevent scFv–GFP depletion, which can cause artifacts[45,96] (Extended Data Fig. 2). Unfertilized embryos were collected from *Vasa-mApple/Matα-GAL4; UAS-suntag-nos (WT or SREmut)/scFv-GFP* virgin female flies that had mated with sterile male flies to produce eggs (male progeny of *osk301/oskCE4* females). Dechorionated embryos were mounted onto the coverslip of a glass-bottom 35 mm dish with their posterior poles pointed towards and glued onto the coverslip (Fig. 3a) to allow the best imaging of germplasm. Live embryos were imaged with a Zeiss LSM980 confocal microscope through the 63× oil objective lens (Plan-Apochromat, 1.4 NA) using AiryScan 2 detector and SR mode. Germplasm was first located and moved into focus using Vasa–mApple through the red fluorescence channel (excitation laser 561 nm) with 1× zoom. Then a small region of interest was imaged (292 pixels × 292 pixels, 12.45 µm × 12.45 µm) and translation sites (bright GFP foci) were identified through the GFP channel (excitation laser 488 nm). Time-lapse images (movies) were acquired with 10 s per frame and 40 frames in total. In each frame, 25 $z$-stacks with a 150 nm interval were imaged with the GFP channel only. For the FRAP experiment, multiple regions containing translation sites were selected and photo-bleached with 70% power 488 nm laser for ten iterations. Three frames were taken before bleaching.

Images were analysed as maximum intensity projections. We tracked individual translation sites with a Fiji plugin, TrackMate (v6.0.2). Laplacian of Gaussian (LoG) detector was used to detect translation sites, with an estimated blob diameter 0.4 µm and sub-pixel localization. Detection thresholds were adjusted for individual images. A simple linear assignment problem (LAP) tracker was used to track the foci movement, with a maximum gap distance of 1 µm and a maximum gap of one frame. Although overall cytoplasmic movement is reduced in unfertilized eggs, translation sites were still undergoing constant and stochastic movement and might move out of the imaging field, which resulted in many short tracks in the tracking result. Therefore, only long tracks (>30 frames, 5 min in total) were selected for further analysis. For the photo-bleached GFP foci, they were usually undetectable by the tracking program for four to six frames before fluorescence recovered to the detection limit. During this period, photo-bleached foci were manually tracked; the tracks of the same spot before bleaching and after fluorescence recovery could be manually identified and connected. The intensities of individual foci in each frame were then extracted from the result file and plotted with time on Prism 8.

### Analysis of FRAP data and calculation of translation elongation rate

The basic assumptions of using FRAP experiments to calculate translation elongation rate have been discussed previously[43,44,64]. We assume that (1) ribosomes are uniformly distributed within the ORF; (2) ribosome elongation rate is constant; (3) scFv-GCN4 epitope binding is stable, and exchange is at a significantly slower rate than translational elongation; (4) nascent peptides are immediately released when synthesis completes.

The first phase of fluorescence recovery after the photo-bleaching is linear due to the synthesis of new SunTags at a constant rate while the fully synthesized and released SunTag-Nanos proteins were still labelled with photo-bleached scFv–GFPs, thus not contributing to the signal change of the polysome. We defined that $L_1$ is the length of the SunTag array and $L_2$ is the length of the Nanos CDS. The linear phase lasts until the fluorescence intensity recovers to $I_0 \times L_2/(0.5L_1 + L_2)$, where $I_0$ is the initial fluorescence intensity before bleaching.

The recovered fluorescence intensity over time is calculated as $I(t) = 2 \times v \times t \times \frac{I_0}{(L_1 + 2L_2)}$ in which $t$ is the time after photo-bleaching and $v$ is the elongation rate.

The second phase starts when the first SunTag synthesized post-photo-bleaching leaves the polysome with fully synthesized protein, which counteracts the increase of newly synthesized SunTag and causes the increase of signal to slow down. When the first ribosome loaded after photo-bleaching finish the translation, the signal reaches a plateau (the third phase) with the same intensity as before photo-bleaching because all the SunTags are bound by non-bleached scFv–GFP again. Indeed, our FRAP curves showed these three phases (Extended Data Fig. 6e). We used the data of the linear phase to fit the linear equation above to calculate the translational elongation rate.

## Calculation of ribosome occupancy

The rationale and mathematical basis of calculating ribosome occupancy using fluorescence of polysomes and single SunTag protein is based on previous studies[40,41,43,44,64,67]. To measure ribosome occupancy, we used rabbit anti-GCN4 antibody to detect the SunTag peptides in embryos from *Vasa-mApple; suntag-nanos* flies, which provides high signal-to-noise ratio and allows clear visualization of single fully synthesized SunTag-Nanos protein (Extended Data Fig. 6a). Fluorescence intensities of polysomes are generally five- to tenfold higher than a single synthesized SunTag-Nanos peptide, so polysomes can be detected in FISH-Quant without detecting single peptides by setting a relatively high threshold. In fact, the automatically assigned threshold in the pre-detection step in FISH-Quant has always been higher than single peptides. To specifically detect single peptides with FISH-Quant, a region in the soma where there are only single peptides (no polysome) is selected, and a low threshold is used for pre-detection (at least tenfold lower than the automatically assigned threshold). In the pre-detection plot, a slope is usually observed at a range of low thresholds before the exponential increase of the detected number at lower thresholds, which corresponds to the background signal. The detection threshold is placed in the middle of the slope, which can capture most of the distinguishable single peptide spots while leaving out the dimer spots that may represent the degrading peptides or peptides not fully labelled. This may also cause an overestimation of the intensity of a single SunTag protein and, consequently, under-estimation of ribosome occupancy. Raw intensities of polysomes and single peptides were extracted from the result file of FISH-Quant analysed and plotted in Prism to obtain mean intensities of each population, where $F$ represents the intensity of a polysome and $F_0$ represents the intensity of a single protein.

As the ribosomes within the SunTag CDS synthesize only part of the SunTag repeats, they do not contribute to the fluorescence as much as the ribosomes within the Nanos CDS, which have the complete SunTag repeats. Assuming ribosomes are uniformly distributed throughout the CDS, the ribosomes within the SunTag repeats on average have half of the SunTag repeats. Therefore, the effective length of the ORF equals $0.5L_1 + L_2$, where $L_2$ is the length of Nanos and $L_1$ is the length of the SunTag. The fluorescence intensity of a polysome $F = F_0 \times d \times (L_2 + 0.5L_1)$, where $d$ is the density of ribosomes in a polysome.

## In vitro egg activation

The protocol is adapted on the basis of previous studies[50,97]. Young (less than 1 week old) *Vasa-mApple/+; suntag-nanos, scFv-GFP/+* flies were well fed to enrich late-stage oocytes, which were then dissected out in 1× Robb's buffer. Stage 14 oocytes were identified and sorted on the basis of the morphology of the dorsal appendages and transferred into 30% Robb's buffer to be incubated and activated for over 30 min, during which oocytes became swollen and some dorsal appendages became separated. Incubated oocytes were incubated in 50% bleach for 1 min, during which non-activated oocytes were lysed by bleach while activated oocytes survived due to vitelline membrane cross-linking and were immediately washed thoroughly with 30% Robb's buffer. Activated oocytes were mounted on the coverslip of a glass-bottom dish with the posterior pole stuck onto the coverslip by heptane glue.

A small piece of wet tissue was put inside the dish to humidify the internal. Oocytes were imaged live with Zeiss LSM980 confocal microscope with the 63× oil lens and Airyscan 2 detector.

## Structure modelling

The predicted structure model of short Oskar was generated using AlphaFold2 (https://github.com/sokrypton/ColabFold/blob/main/AlphaFold2.ipynb)[98,99]. The structures of the Lotus and SGNH domains have been previously characterized directly using X-ray crystallography and correspond well with the AlphaFold prediction[37,39].

## Oskar sequence feature analysis

The sequences of the 11 *Drosophila* species used (*D. melanogaster* (NP_996186.1), *D. immigrans* (KAH8311831.1), *D. virilis* (XP_002053269.1), *D. hydei* (XP_023173869.2), *D. miranda* (XP_017140399.1), *D. grimshawi* (XP_001994345.1), *D. navojoa* (XP_017968973.1), *D. pseudoobscura* (XP_001359508.2), *D. arizonae* (XP_017856611.1) and *D. persimilis* (XP_002017385.1)) were aligned and conservation plots were acquired in Benchling. Disorder sequence prediction was performed on the IUPred2A website (https://iupred2a.elte.hu/)[100].

## Quantification of translation in germplasm induced by *Oskar-NQmut*

To compare the *suntag-nanos* translation on Oskar-NQmut germ granules with wild-type Oskar, we generated flies expressing *suntag-nanos* or *suntag-nanos-SREmut*, together with *UAS-Oskar-WT/NQmut-bcd* 3' UTR transgene to induce germ granules at the anterior pole of the embryos (see detailed genotypes in Supplementary Table 1). Embryos were collected, fixed and stained with *suntag* smFISH probes and anti-GCN4. Images were acquired from the induced germplasm at the anterior pole as well as the native germplasm at the posterior. The percentage of translating mRNA was measured using FISH-Quant using the quantification protocol described above. The translation activity at the anterior of an embryo is measured by the translating fraction of the anterior germplasm normalized with the fraction of the posterior germplasm.

## FRAP of germ granules

To assess the dynamics of germ granules made by Oskar-WT or Oskar-NQmut, embryos expressing *Vasa-mApple* and *Oskar-WT-bcd* 3' UTR or *Oskar-NQmut-bcd* 3' UTR were collected, mounted with anterior poles on a coverslip and imaged live with Zeiss LSM780 with red fluorescence channel (excitation wavelength 561 nm). An area of about 5 μm × 5 μm size was chosen in germplasm and photo-bleached with 70% 561 nm laser. The fluorescence of the bleached region was recorded with time-lapse imaging (5 s interval), measured as integrated intensity using Fiji and plotted over time using Prism 8.

## Cuticle preparation and imaging

Embryos were collected overnight from the *Matα-GAL4VP16/UAS-osk (WT or NQmut)-bcd* 3' UTR flies and aged for 24 h, after which embryos were dechorionated with 50% bleach for 2 min, extensively washed and then transferred to a mesh-bottom basket. The embryos in the basket were incubated with the acidic acid/glycerol 4:1 mixture for 1 h at 60 °C, after which embryos were transferred to a slide, covered with Hoyer's medium and coverslip, and incubated overnight at 60 °C. The cuticles were examined by dark-field microscopy.

## Statistics and reproducibility

No statistical methods were used to pre-determine sample sizes. We performed quantification (for translating fractions, elongation rates, spot distribution, FRAP and so on) with sample sizes similar to those reported in previous publications[40,41,43,69,81]. Samples from wild-type (control) and mutant (experimental) fly strains were equally and randomly collected. No data were excluded from analyses.

The investigators were not blinded to allocation during experiments and outcome assessment. When Student's *t*-test (two-tailed, at 95% confidence intervals) was performed, data distribution was assumed to be normal, but this was not formally tested. All quantitative results have been repeated with similar results at least once independently. The microscopic images shown in the figures are all representative images of more than three similar images acquired from one experiment and have been reproduced in another independent experiment.

## Reporting summary

Further information on research design is available in the Nature Portfolio Reporting Summary linked to this article.

## Data availability

Previously published Oskar protein sequences from different *Drosophila* species that were reanalysed here are available under accession codes NP_996186.1, KAH8311831.1, XP_002053269.1, XP_023173869.2, XP_017140399.1, XP_001994345.1, XP_017968973.1, XP_001359508.2, XP_017856611.1 and XP_002017385.1. Source data are provided within this paper. All other data supporting the findings of this study are available within the manuscript or can be obtained from the corresponding author on reasonable request.

## Code availability

Custom Python scripts and ImageJ macro used to measure the distance between foci signals to granule borders have been uploaded to GitHub (https://github.com/wstainier/mRNA_distance_measurements).

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

## Acknowledgements

We thank M. Pamula, S. Grill, B. Lin and J. Rajakumar for their constructive and critical feedback on the work and manuscript; E. Dawson for the initial conceptualization of the project. We thank P. Lasko and E. Wahle for sharing antibodies and G. Gonsalvez for the Oskar–GFP fly stock. We thank the Drosophila Genomics Resource Center for reagents, Bloomington Drosophila Stock Center and Kyoto Drosophila Stock Center for fly stocks. This work was partially supported by Howard Hughes Medical Institute to R.L. Part of this work was initially supported by an HFSP grant to M.L. and ANR MemoRNP. M.L. and J.D. are supported by the CNRS.

## Author contributions

R.C. and R.L. designed the experiments. R.C., W.S. and J.D. performed the investigation. R.C. wrote the original draft. All authors reviewed and edited the text. R.L. and M.L. acquired the funding. R.L. supervised the work.

## Competing interests

The authors declare no competing interests.

## Additional information

**Extended data** is available for this paper at https://doi.org/10.1038/s41556-024-01452-5.

**Correspondence and requests for materials** should be addressed to Ruth Lehmann.

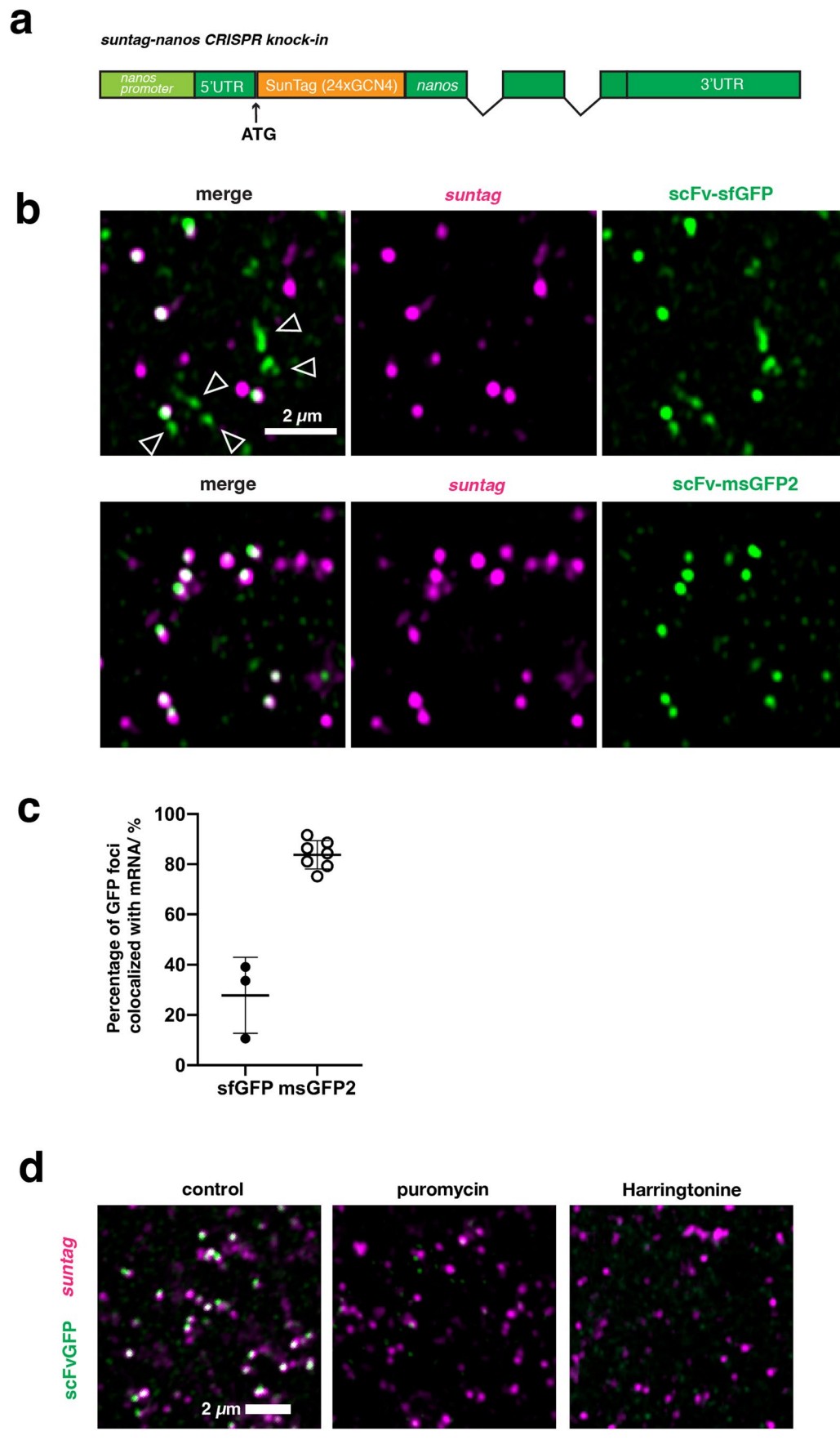

**Extended Data Fig. 1 | See next page for caption.**

**Extended Data Fig. 1 | Optimization and validation of the *suntag-nanos* system. a**, Schematic of CRISPR knocked-in *suntag-nanos* allele. **b**, Images of germplasm in embryos expressing *suntag-nanos* and (top) scFv-sfGFP (super-folder GFP) or (bottom) monomeric msGFP2 (green). *Suntag* mRNA is stained by *suntag* probes (magenta). ScFv-sfGFP showed puncta of GFP signals (arrowheads) which are not co-localized with mRNA signal and thus are not translating sites. ScFv-msGFP2, which is used throughout this study, strongly reduces the aggregation. **c**, The percentage of bright GFP foci co-localized with mRNA. Large aggregates of scFv-sfGFP colocalize rarely with RNA (-30%) (n = 3 embryos). With scFv-msGFP2, the majority of GFP foci are associated with RNA (80%-90%) and most likely represent polysomes (n = 7 embryos). Data are the mean ± s.d. **d**, Embryos expressing *suntag-nanos* and scFv-GFP (green) treated with 20 mM HEPES (control), 10 mg/ml puromycin, or 100 µg/ml harringtonine. Treated embryos were aged for at least 15 min before fixation and staining with *suntag* probes (magenta).

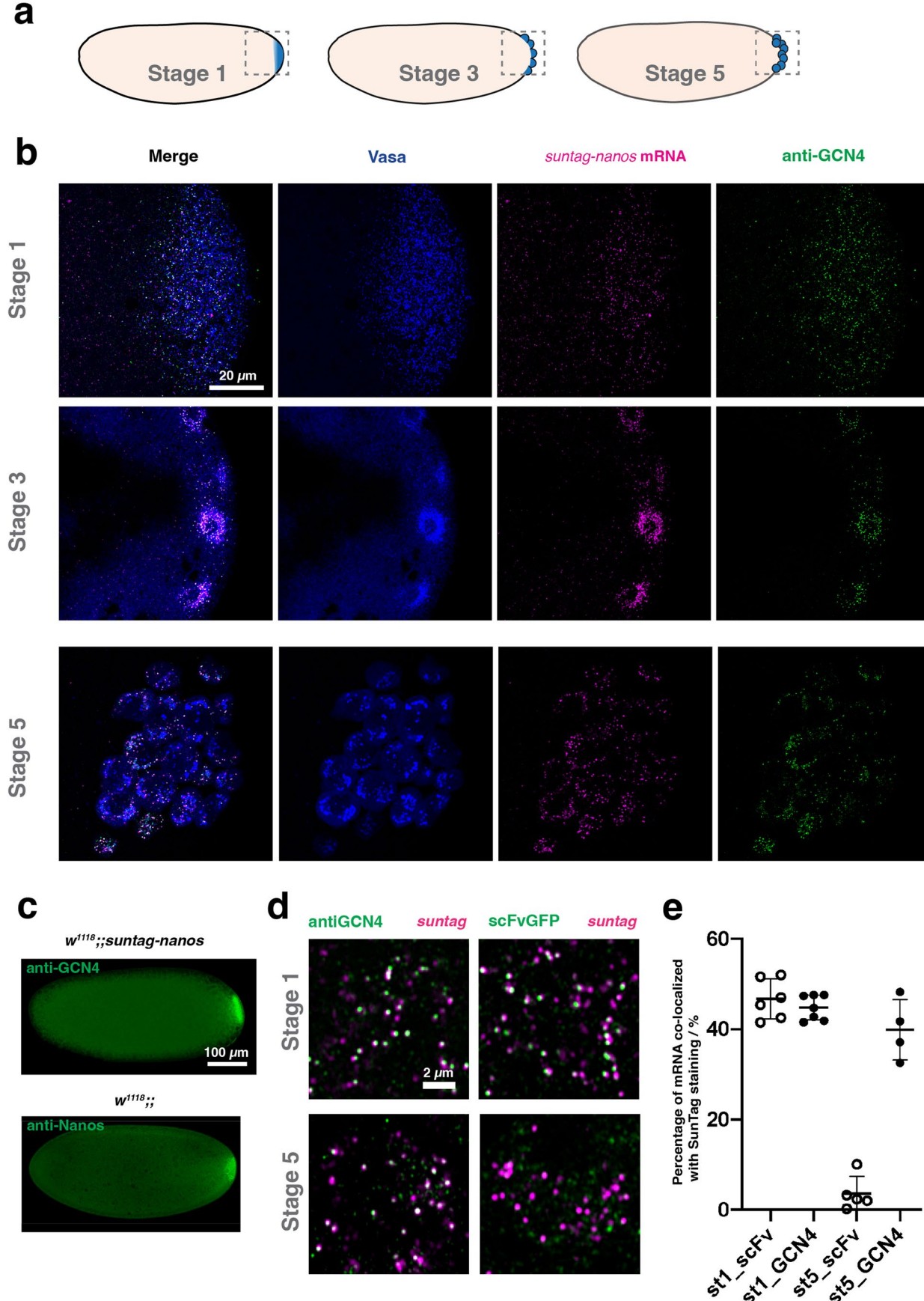

**Extended Data Fig. 2 | See next page for caption.**

**Extended Data Fig. 2 | Quantitative detection of SunTag with anti-GCN4 versus scFv-GFP. a**, Schematics of stage-1, stage-3, and stage-5 embryos. Germplasm or pole cells are labeled in blue. Outlined regions are imaged and presented in panel (B). **b**, Example images of stage-1, stage-3, and stage-5 embryos expressing *suntag-nanos*. SunTag is stained by anti-GCN4 (green); suntag mRNA is stained by smFISH (magenta); germplasm or pole cells are marked by Vasa-mApple (blue). **c**, Images of a stage-1 embryo expressing *suntag-nanos* stained by anti-GCN4 (top) and a stage-1 embryo from a *w^{1118}* female stained by anti-Nanos (bottom). **d**, Images of germplasm in stage-1 embryos (top) and pole cells in stage-5 embryos expressing *suntag-nanos*. SunTag (green) is stained by anti-GCN4 (left) or by endogenous scFv-GFP (right); *suntag* mRNA is stained by smFISH (magenta). **e**, Quantification of the percentage of mRNA foci co-localized with SunTag staining signal in stage-1 germplasm and stage-5 pole cells when SunTag is stained by scFv-GFP or anti-GCN4. $n_{st1-scFv} = 6$, $n_{st1-GCN4} = 7$, $n_{st5-scFv} = 5$, $n_{st5-GCN4} = 4$; n refers to the number of embryos. Data are the mean ± s.d.

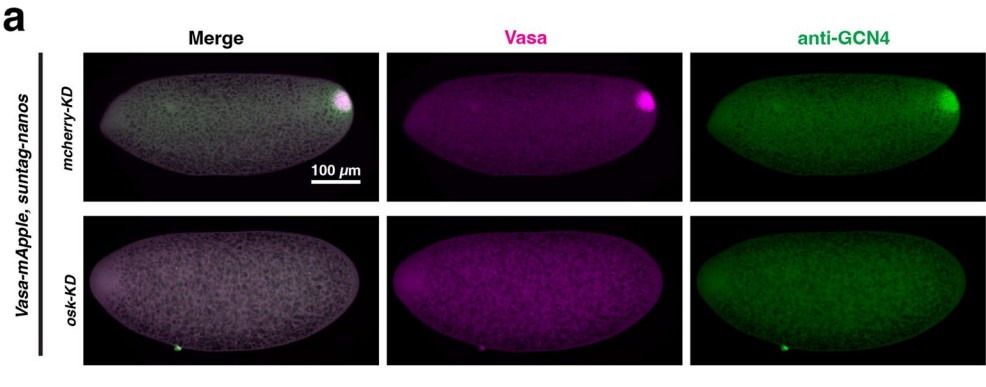

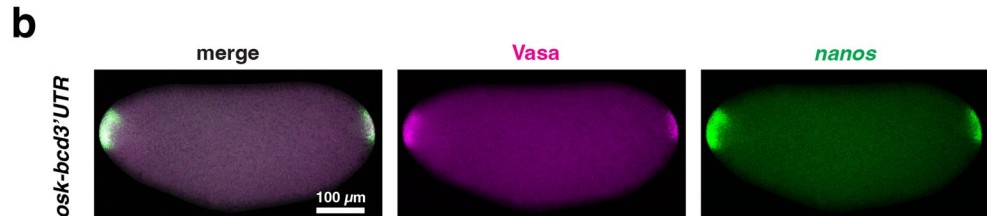

**Extended Data Fig. 3 | *Suntag-nanos* mRNA translation depends on germplasm. a**, Images of embryos expressing *suntag-nanos* with *mcherry* (control) knockdown (top) or *osk* knockdown (bottom). SunTag is stained by anti-GCN4 (green) and germplasm is marked by Vasa-mApple (magenta). **b**, Images of embryos expressing Vasa-mApple and *osk-bcd*3'*UTR*, forming germplasm and localizing *nanos* mRNA at the anterior pole. Germplasm is marked by Vasa-mApple (magenta). Endogenous *nanos* mRNA is stained by smFISH probes against *nanos* (green).

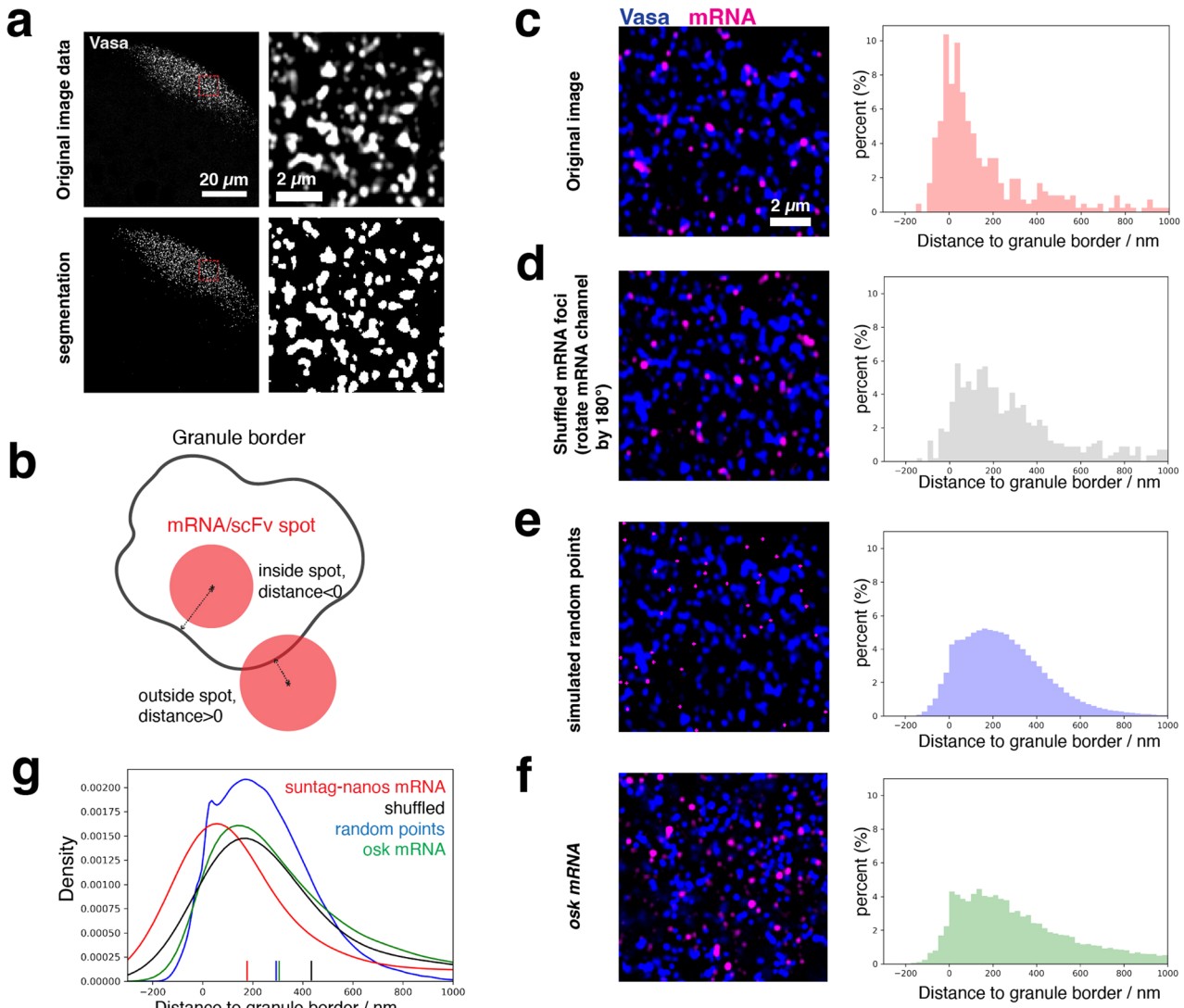

**Extended Data Fig. 4 | Granule segmentation and distance measurement.**
**a**, Images of germ granules are segmented with the ilastik program. The top shows the original grayscale images of Vasa-mApple at the posterior pole of an embryo (left) and zoomed image of the outline region in the germplasm (right). The bottom shows the black-and-white binary images of segmented germ granules (granules in white). **b**, Schematic of the distance measurement program. The granule surface is defined after segmentation by ilastik. The coordinates of mRNA smFISH or scFv-GFP/anti-GCN4 spots are determined by FISH-Quant and used to measure the distance to the closest granule surface. The schematic is drawn in 2D but the actual data and measurement are in 3D. **c-f**, Control and validation experiments of distance measurement. (Left) representative images of germplasm with germ granules marked by Vasa-mApple (blue). Magenta: **c** suntag mRNA smFISH; **d** is the same image as **c** with mRNA channel rotated by 180° to shuffle the mRNA distribution; **e** has the same Vasa

channel image as **c** with one million simulated points randomly distributed within the image; **f** smFISH of *osk* mRNA, which is not a component of germ granules. The distributions of mRNA foci or simulated points are plotted in the relative frequency histograms on the right. The x-axis refers to the distance of foci centroids to the border of the closest granule; the zero marks the granule border; a negative value denotes being inside a granule and a positive value denotes outside. Numbers of foci plotted: c, 415 foci from one embryo; d, 563 foci from one embryo; e, 249672 foci from one embryo; f, 8078 foci from four embryos. **g**, The distributions of mRNA foci or simulated points in (**c-f**) are plotted together as a kernel density estimate (KDE) plot. The distributions of shuffled mRNA, random points, and *osk* mRNA show a shift away from the surface of germ granules when compared to *suntag-nanos* mRNA. The ticks on the x-axis represent the mean values of the distributions.

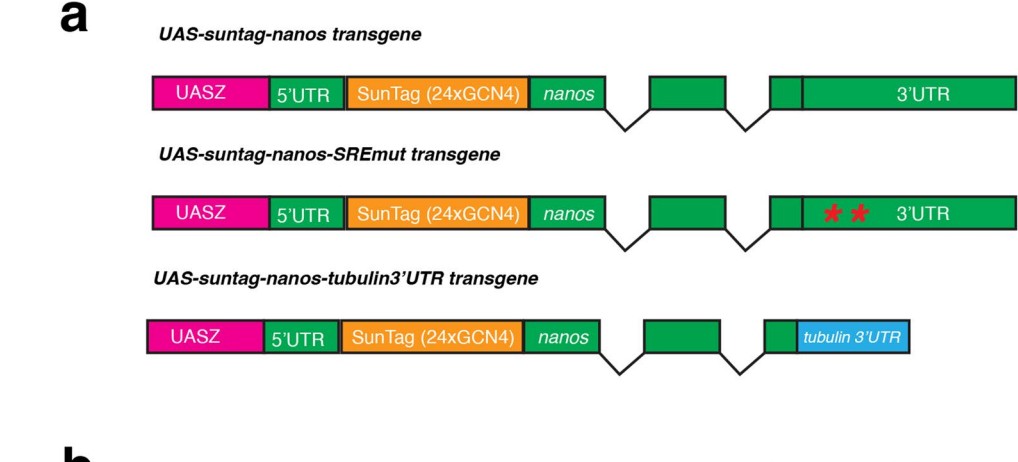

**b**

| Merge | Vasa | *suntag-nanos* mRNA | anti-GCN4 |

*UAS-suntag-nanos transgene*

*UAS-suntag-nanos-SREmut transgene*

*UAS-suntag-nanos-tub3'UTR transgene*

20 µm

**Extended Data Fig. 5 | Translation regulation of *suntag-nanos* is mediated by *nanos* 3'UTR. a**, Schematics of transgenic constructs of *UAS-suntag-nanos*, *UAS-suntag-nanos-SREmut*, and *UAS-suntag-nanos-tubulin*3'*UTR*. The red asterisks in *nanos* 3'UTR represent the two SREs mutated in the construct. **b**, Representative images of embryos expressing *suntag-nanos* (top), *suntag-nanos-SREmut* (middle), and *suntag-nanos-tubulin*3'*UTR* (bottom). Note that the translation activities in the soma of embryos expressing *suntag-nanos-SREmut* and *suntag-nanos-tubulin*3'*UTR* are higher than *suntag-nanos*. Blue, Vasa; magenta, suntag smFISH; green, anti-GCN4.

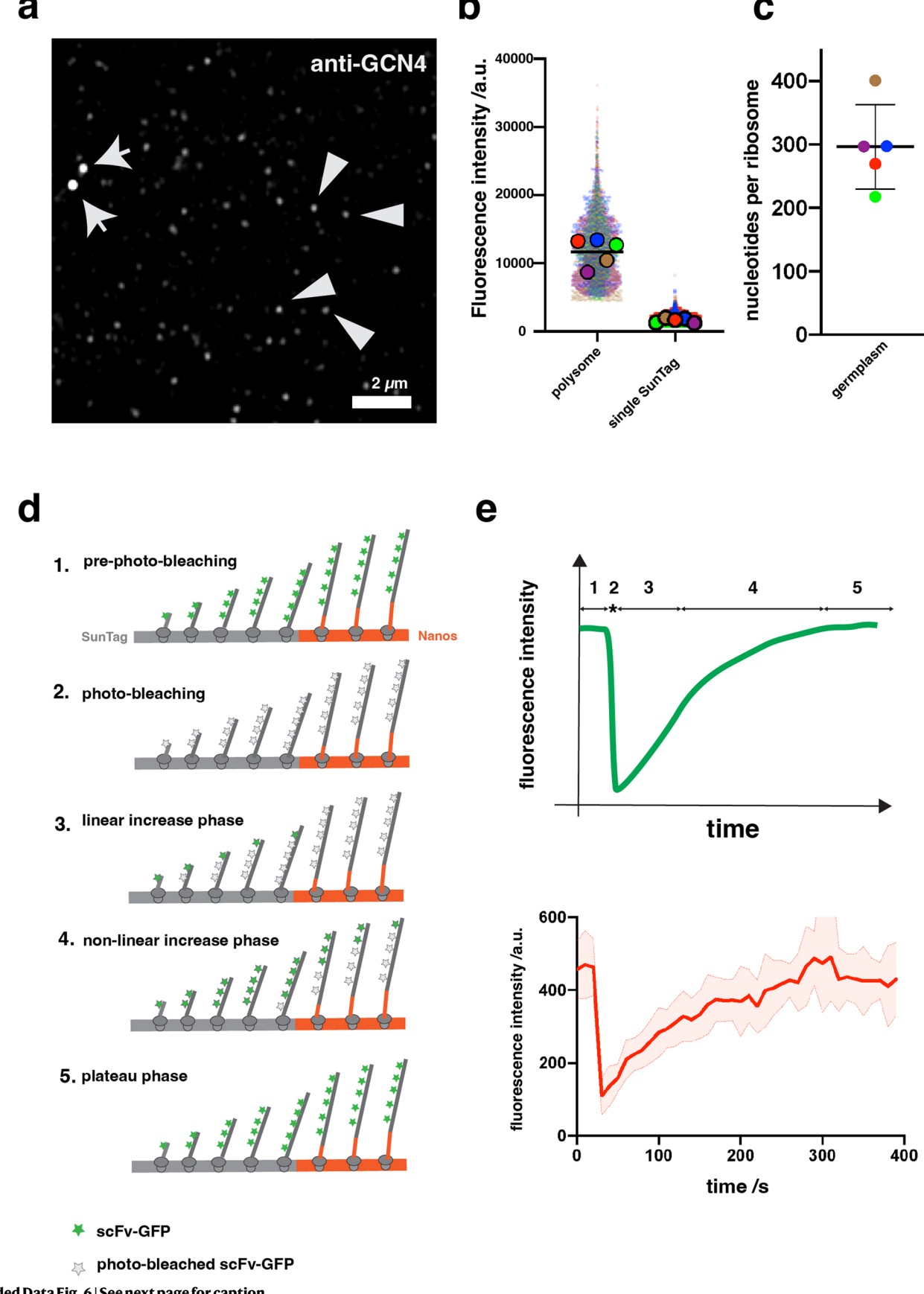

**Extended Data Fig. 6 | See next page for caption.**

**Extended Data Fig. 6 | Quantification of the intensity of polysomes, ribosome occupancy, and translation elongation. a**, Germplasm of an embryo expressing *suntag-nanos* flies with SunTag detected by anti-GCN4. The polysomes (arrows) have stronger fluorescence intensities and are co-localized with the mRNA signal (not shown). Individual fully synthesized SunTag-Nanos proteins (examples pointed out by arrowheads) have lower intensities and are not co-localized with mRNA. **b**, Fluorescence intensities of polysomes and single SunTag-Nanos protein, extracted from FISH-Quant analysis (see Methods). Data from five embryos, each represented by a different color, are plotted as a super-plot. Color-filled circles represent the mean values of the embryos. **c**, Calculated ribosome occupancy on *suntag-nanos* mRNA using data from **b**. Data are the mean ± s.d. **d**, Theoretical process of fluorescence recovery after photo-bleaching (FRAP). Before photo-bleaching, *suntag-nanos* mRNA is translated at a steady state with SunTag bound by fluorescent scFv-GFP (phase 1). Photo-bleaching diminishes the fluorescence of bound scFv-GFP (phase 2). Newly synthesized SunTag epitopes after photo-bleaching bind fluorescent scFv-GFP, causing fluorescence recovery of the polysome. Assuming a constant elongation rate, the initial phase of recovery is linear (phase 3). When the peptide that contains the first SunTag synthesized post-bleaching leaves polysome, which counteracts the increase of newly synthesized SunTag, the increase of signal starts to slow down (phase 4). When the first ribosome loaded after photo-bleaching finishes the translation, the signal reaches a plateau (phase 5) with the same intensity as before photo-bleaching because all the SunTags are bound by fluorescent scFv-GFP again. **e**, A hypothetical FRAP curve (top) based on the theoretical FRAP process, and the FRAP experimental data (bottom, same as Fig. 3e, Data are the mean ± s.d.), which shows a similar curve as the theoretical curve.

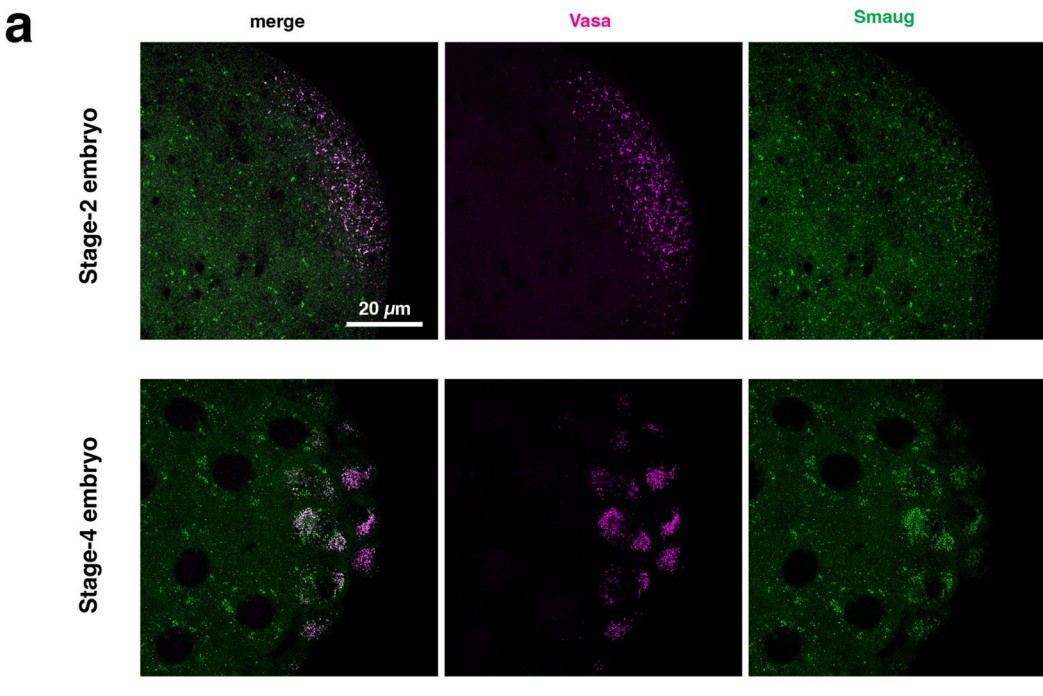

**Extended Data Fig. 7 | See next page for caption.**

**Extended Data Fig. 7 | Distribution of Smaug and ME31B. a**, Stage-2 (top) and stage-4 (bottom) embryos expressing Vasa-mApple and Smaug-GFP, showing the morphology and distribution of Smaug (green) in soma and germplasm. In the soma, Smaug forms heterogeneous puncta. In germplasm, Smaug is enriched in germ granules (magenta). **b**, Stage-2 (top) and stage-4 (bottom) embryos expressing Vasa-mApple and ME31B-GFP, showing the distribution of ME31B (green). At stage 2, ME31B is homogeneously distributed throughout the embryo. At stage 4 and later, ME31B forms large and heterogeneous clusters in the soma and forms small clusters associated with germ granules (magenta) in pole cells, as shown in the zoomed image of the outlined area. The averaged line scan of five granules from 2 embryos for each stage was shown. Data are the mean ± s.d.

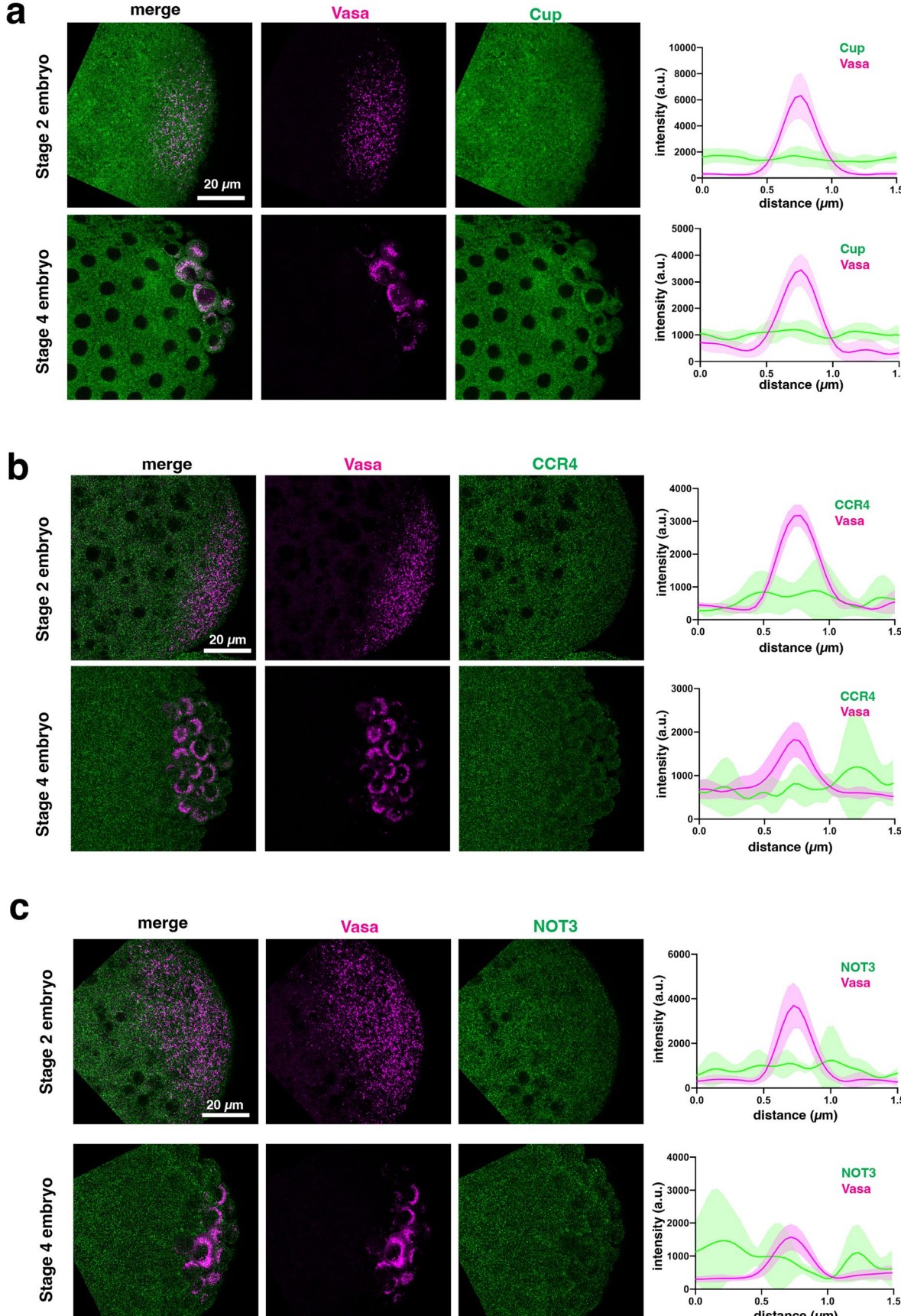

**Extended Data Fig. 8 | Distribution of Cup, CCR4, and NOT3. a**, Embryos expressing Cup-YFP (green) and Vasa-mCherry (magenta). **b**, Embryos expressing Vasa-mApple (magenta) stained with anti-CCR4 antibody (green). **c**, Embryos expressing Vasa-mApple (magenta) stained with anti-NOT3 antibody (green). Stage-2 embryos are shown on the top and stage-4 embryos are shown at the bottom. For each examined protein (Cup/CCR4/NOT3), the averaged line scan of five germ granules from 2 embryos for each stage was shown. Data are the mean ± s.d.

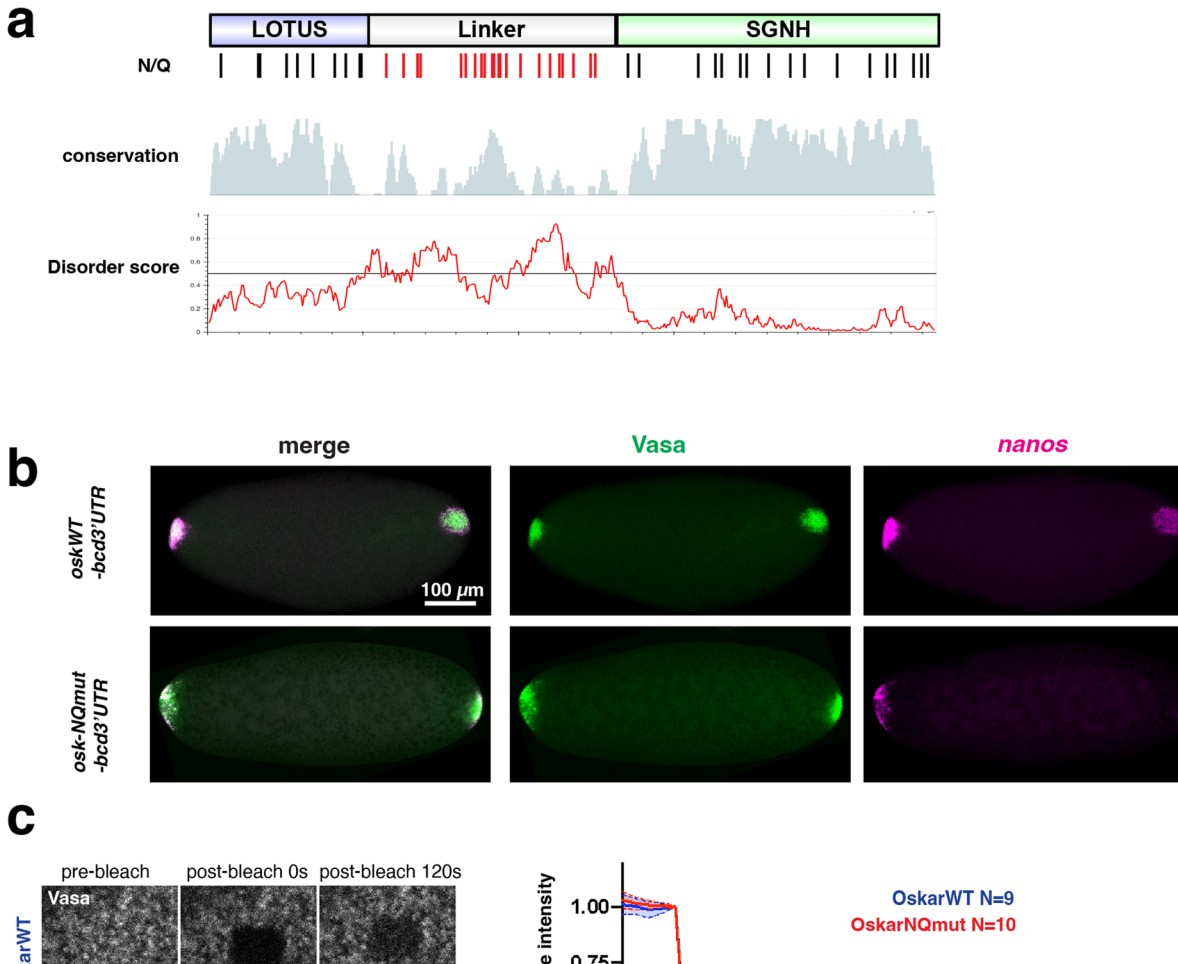

**Extended Data Fig. 9 | Characterization of Oskar-NQmut. a**, Sequence features of short Oskar protein. (Top to bottom) The first track shows the domain structure of Oskar. The second track shows the distribution of Asparagine (N) and Glutamine (Q) residues in the Oskar protein of *Drosophila melanogaster*. The third track shows the sequence conservation of Oskar proteins of 11 Drosophila species. The fourth track shows disorder prediction of the Oskar sequence using IUPred2A online tool. **b**, Embryos expressing *oskWT-bcd3′UTR*
(top) or *osk-NQmut-bcd3′UTR* (bottom). Germplasm is marked by Vasa (green) and *nanos* mRNA is stained by smFISH (magenta). **c**, FRAP of Vasa-mApple in anterior germplasm of embryos expressing *oskWT-bcd3′UTR* (top) or *osk-NQmut-bcd3′UTR* (bottom). Fluorescence intensity over time (WT: blue; NQmut: red) is plotted on the right. N refers to the number of embryos where measurements were made. Data are the mean ± s.d.

| | Corresponding author(s): | Ruth Lehmann |
|---|---|---|
| | Last updated by author(s): | May 20, 2024 |

# Reporting Summary

## Statistics

For all statistical analyses, confirm that the following items are present in the figure legend, table legend, main text, or Methods section.

| n/a | Confirmed | |
|---|---|---|
| ☐ | ☒ | The exact sample size (*n*) for each experimental group/condition, given as a discrete number and unit of measurement |
| ☐ | ☒ | A statement on whether measurements were taken from distinct samples or whether the same sample was measured repeatedly |
| ☐ | ☒ | The statistical test(s) used AND whether they are one- or two-sided *Only common tests should be described solely by name; describe more complex techniques in the Methods section.* |
| ☒ | ☐ | A description of all covariates tested |
| ☐ | ☒ | A description of any assumptions or corrections, such as tests of normality and adjustment for multiple comparisons |
| ☐ | ☒ | A full description of the statistical parameters including central tendency (e.g. means) or other basic estimates (e.g. regression coefficient) AND variation (e.g. standard deviation) or associated estimates of uncertainty (e.g. confidence intervals) |
| ☐ | ☒ | For null hypothesis testing, the test statistic (e.g. *F*, *t*, *r*) with confidence intervals, effect sizes, degrees of freedom and *P* value noted *Give P values as exact values whenever suitable.* |
| ☒ | ☐ | For Bayesian analysis, information on the choice of priors and Markov chain Monte Carlo settings |
| ☒ | ☐ | For hierarchical and complex designs, identification of the appropriate level for tests and full reporting of outcomes |
| ☐ | ☒ | Estimates of effect sizes (e.g. Cohen's *d*, Pearson's *r*), indicating how they were calculated |

*Our web collection on statistics for biologists contains articles on many of the points above.*

## Software and code

Policy information about availability of computer code

| Data collection | Zeiss ZEN 3.8 |
|---|---|
| Data analysis | MatLab_R2021a, FIJI (ImageJ Version 2.14.0), ilastik (1.4.0b27), FISHQuant_v3, Colabfold (v1.5.2), IUPred2A, GraphPad Prism (8.4.3), Python3 and custom code run on python3 for image analysis. BioRender was used for creating figures. Custom Python scripts and ImageJ macro used to measure the distance between foci signals to granule borders have been uploaded to GitHub: https://github.com/wstainier/mRNA_distance_measurements. |

For manuscripts utilizing custom algorithms or software that are central to the research but not yet described in published literature, software must be made available to editors and reviewers. We strongly encourage code deposition in a community repository (e.g. GitHub). See the Nature Portfolio guidelines for submitting code & software for further information.

## Data

Policy information about availability of data

All manuscripts must include a data availability statement. This statement should provide the following information, where applicable:

- Accession codes, unique identifiers, or web links for publicly available datasets
- A description of any restrictions on data availability
- For clinical datasets or third party data, please ensure that the statement adheres to our policy

Previously published Oskar protein sequences from different Drosophila species that were re-analyzed here are available under accession codes NP_996186.1,

## Research involving human participants, their data, or biological material

Policy information about studies with human participants or human data. See also policy information about sex, gender (identity/presentation), and sexual orientation and race, ethnicity and racism.

| | |
|---|---|
| Reporting on sex and gender | No human data in this study. |
| Reporting on race, ethnicity, or other socially relevant groupings | No human data in this study. |
| Population characteristics | No human data in this study. |
| Recruitment | No human data in this study. |
| Ethics oversight | No human data in this study. |

Note that full information on the approval of the study protocol must also be provided in the manuscript.

# Field-specific reporting

Please select the one below that is the best fit for your research. If you are not sure, read the appropriate sections before making your selection.

☒ Life sciences  ☐ Behavioural & social sciences  ☐ Ecological, evolutionary & environmental sciences

For a reference copy of the document with all sections, see nature.com/documents/nr-reporting-summary-flat.pdf

# Life sciences study design

All studies must disclose on these points even when the disclosure is negative.

| | |
|---|---|
| Sample size | No sample size calculation was performed. For the quantification of translation, 5-10 embryos were used for each genotype or condition; for the distance measurement, results from at least three embryos were combined; which we found sufficient to demonstrate the robustness and variability of the results based on preliminary experiments. |
| Data exclusions | No data has been systematically excluded. |
| Replication | All quantification experiments have been repeated at least once independently with similar results. |
| Randomization | Samples from wildtype (control) and mutant (experimental) fly strains were equally and randomly collected. |
| Blinding | Blinding was not done. Blinding was not possible when different groups can be easily identified during data collection (germplasm vs soma difference, wildtype vs mutant difference). Blinding was not necessay or practical during data analysis because all analysis was done using software automatically with the same setting. |

# Reporting for specific materials, systems and methods

We require information from authors about some types of materials, experimental systems and methods used in many studies. Here, indicate whether each material, system or method listed is relevant to your study. If you are not sure if a list item applies to your research, read the appropriate section before selecting a response.

## Materials & experimental systems

| n/a | Involved in the study |
|---|---|
| ☐ | ☒ Antibodies |
| ☒ | ☐ Eukaryotic cell lines |
| ☒ | ☐ Palaeontology and archaeology |
| ☐ | ☒ Animals and other organisms |
| ☒ | ☐ Clinical data |
| ☒ | ☐ Dual use research of concern |
| ☒ | ☐ Plants |

## Methods

| n/a | Involved in the study |
|---|---|
| ☒ | ☐ ChIP-seq |
| ☒ | ☐ Flow cytometry |
| ☒ | ☐ MRI-based neuroimaging |

# Antibodies

| | |
|---|---|
| Antibodies used | Commercial antibodies:rabbit anti-RPS6 (Cell Signaling #2217), rabbit anti-GCN4 (Novus Bio #C11L34 1:1000),ThermoFisher Scientific anti-rabbit AlexaFluor488 (A-11008), anti-rabbit AlexaFluor647 (A-31573) , anti-rat AlexaFluor555 (A-21434).<br>Lab-generated primary antibodies: rat anti-Oskar (gift from Paul Lasko) , rabbit anti-Nanos (Lehmann Lab), rabbit anti-CCR4 and anti-NOT3 (gift from Elmar Wahle) |
| Validation | Validations of commercial antibodies are described on manufacturers' websites:<br>rabbit anti-RPS6:https://www.cellsignal.com/products/primary-antibodies/s6-ribosomal-protein-5g10-rabbit-mab/2217?_requestid=3689689<br>rabbit anti-GCN4:https://www.novusbio.com/products/gcn4-antibody-c11l34_nbp2-81274<br>For anti-Oskar and anti-Nanos, we validated them by showing that the embryo staining results is consistent with the published localization of the proteins (Ephrussi and Lehmann, Nature. 1992 Jul 30;358(6385):387-92). Anti-CCR4 and anti-NOT3 have been validated by the published study: Temme et. al., RNA. 2010 Jul; 16(7): 1356–1370. |

# Animals and other research organisms

Policy information about <u>studies involving animals</u>; <u>ARRIVE guidelines</u> recommended for reporting animal research, and <u>Sex and Gender in Research</u>

| | |
|---|---|
| Laboratory animals | Species: Drosophila melangaster; strains: w1118, yw, and derived lines. Age: 0-3h embryos. |
| Wild animals | The study did not involve wild animals. |
| Reporting on sex | The samples were eggs and ovaries collected from female flies. The sex of eggs were irrelevant and therefore not determined because processes we studied happen before the sex differentiation of the embryos. |
| Field-collected samples | The study did not invlove samples collected from the field. |
| Ethics oversight | No ethical oversight relevant. |

Note that full information on the approval of the study protocol must also be provided in the manuscript.

