## [Peer Review File · Nature Cell Biology]

Peer Review Information

Journal: Nature Cell Biology

Manuscript Title: Direct Observation of Translational Activation by a Ribonucleoprotein Granule

Corresponding author name(s): Professor Ruth Lehmann

Editorial Notes:

Reviewer Comments & Decisions:

Decision Letter, initial version:
--

*Please delete the link to your author homepage if you wish to forward this email to co-authors.

Dear Professor Lehmann,

Apologies once again for the delay. Your manuscript, "Repressor sequestration activates translation of germ granule localized mRNA", has now been seen by 3 referees, who are experts in *Drosophila* egg chamber (referee 1); biomolecular condensation (referee 2); and translation and RNA imaging (referee 3). As you will see from their comments (attached below) they find this work of potential interest, but have raised substantial concerns, which in our view would need to be addressed with considerable revisions before we can consider publication in Nature Cell Biology.

Nature Cell Biology editors discuss the referee reports in detail within the editorial team, including the chief editor, to identify key referee points that should be addressed with priority, and requests that are overruled as being beyond the scope of the current study. To guide the scope of the revisions, I have listed these points below. We are committed to providing a fair and constructive peer-review process, so please feel free to contact me if you would like to discuss any of the referee comments further.

In particular, it would be essential to:

- A) Experimentally further test effects on translation (Reviewer #2) including further testing of translation inhibition (Reviewer #3)
- B) Test whether and how the SunTagging may affect results (all Reviewers)
- C) Further assess colocalization of nanos and germ granules (all Reviewers) as well as their dynamic

interactions (Reviewer #3).

D) All other referee concerns pertaining to strengthening existing data, providing controls, methodological details, clarifications and textual changes, should also be addressed.

F) Finally please pay close attention to our guidelines on statistical and methodological reporting (listed below) as failure to do so may delay the reconsideration of the revised manuscript. In particular please provide:

We would be happy to consider a revised manuscript that would satisfactorily address these points, unless a similar paper is published elsewhere, or is accepted for publication in Nature Cell Biology in the meantime.

- ensure that it conforms to our format instructions and publication policies (see below and <https://www.nature.com/nature/for-authors>).

- provide a point-by-point rebuttal to the full referee reports verbatim, as provided at the end of this letter.

- provide the completed Reporting Summary (found here <https://www.nature.com/documents/nr-reporting-summary.pdf>). This is essential for reconsideration of the manuscript will be available to editors and referees in the event of peer review. For more information see <http://www.nature.com/authors/policies/availability.html> or contact me.

When submitting the revised version of your manuscript, please pay close attention to our [href="https://www.nature.com/nature-portfolio/editorial-policies/image-integrity">Digital Image Integrity Guidelines](https://www.nature.com/nature-portfolio/editorial-policies/image-integrity). and to the following points below:

Finally, please ensure that you retain unprocessed data and metadata files after publication, ideally

archiving data in perpetuity, as these may be requested during the peer review and production process or after publication if any issues arise.

Nature Cell Biology is committed to improving transparency in authorship. As part of our efforts in this direction, we are now requesting that all authors identified as 'corresponding author' on published papers create and link their Open Researcher and Contributor Identifier (ORCID) with their account on the Manuscript Tracking System (MTS), prior to acceptance. ORCID helps the scientific community achieve unambiguous attribution of all scholarly contributions. You can create and link your ORCID from the home page of the MTS by clicking on 'Modify my Springer Nature account'. For more information please visit www.springernature.com/orcid.

This journal strongly supports public availability of data. Please place the data used in your paper into a public data repository, or alternatively, present the data as Supplementary Information. If data can only be shared on request, please explain why in your Data Availability Statement, and also in the correspondence with your editor. Please note that for some data types, deposition in a public repository is mandatory - more information on our data deposition policies and available repositories appears below.

[Redacted]

We would like to receive a revised submission within six months.

We hope that you will find our referees' comments, and editorial guidance helpful. Please do not hesitate to contact me if there is anything you would like to discuss.

Best wishes,

Daryl

Daryl Jason Verzosa David, PhD

Senior Editor, Nature Cell Biology
Nature Portfolio

Heidelberger Platz 3, 14197 Berlin, Germany
Email: daryl.david@nature.com
ORCID: <https://orcid.org/0000-0002-9253-4805>

Reviewers' Comments:

Reviewer #1:

Remarks to the Author:

A. Summary of the key results

The manuscript by Chen et al uses quantitative single-molecule imaging to make a definitive case that germ granules are the sites of translation of nanos (nos) mRNA in the early *Drosophila* embryo. It further shows that nanos mRNA is oriented within germ granules such that its ORF is outside or on the surface of the granules, while its 3' UTR is embedded within. Surprisingly, Smaug (Smg), a well-established translational repressor of nos mRNA, also accumulates within germ granules. Using a novel separation-of-function allele of oskar (osk) in which a predicted disordered region is altered, the authors showed that sequestration of Smg is dependent on Osk, and in this mutant nos translation is greatly reduced. Within germ granules Smg does not act as a translational repressor, because necessary co-factors such as Cup and the CCR/NOT complex do not accumulate there, but outside of germ granules Smg is active. Thus Osk controls nos translation by enabling Smg to accumulate into a region of the germ granules where it is inactive.

B. Originality and significance

The experiments involve original technologies. In my view these results are highly significant, not just because they answer a long-standing question in the field, but also because it is probable that similar mechanisms function in other ribonucleoprotein particles (RNPs) that form condensates. Such RNPs are ubiquitous in evolution and they operate in numerous cellular and developmental processes.

C. Data and methodology

Overall, the experiments in this manuscript are carefully performed and well documented. Appropriate controls are included.

D. Statistics and uncertainties

The use of statistics is thorough and appropriate, with uncertainties well documented except for point 1 below.

E. Conclusions

The conclusions are robustly supported by the data.

F. Suggested improvements

Because of the high quality of the paper I have only a few relatively minor comments that I would like the authors to address in revision.

1. In the legend to Extended Data Fig. 5, it is noted that measurements around 0 are

underrepresented in all experiments, and that this likely results from an artefact caused by the design of the distance measuring program. Can this be further explained, and how do the authors know that additional artefacts do not further distort the data? Better yet, can it be fixed?

2. In the Supplementary Notes, it is stated that *suntag-nanos* homozygotes are viable and females lay normal numbers of eggs, but the eggs do not hatch. The explanation given is that this phenotype results from low expression of *suntag-nanos* as compared with the endogenous gene. The phenotype of the embryos produced in the eggs should be described. Do they have posterior patterning defects consistent with a *nos* hypomorph, or is their phenotype different and/or more complex?

3. Also relevant to the Supplementary Notes, given the failure to recover one, can the authors speculate as to whether a knock-in with the insertion as designed might cause dominant lethality or dominant sterility?

4. In lines 231-232 there is speculation that recruitment of translation factors to germ granules may facilitate mRNA translation and complement the derepression mechanism that is the major focus of the paper. Extended Data Fig. 9 indeed shows that PABP and eIF4G are enriched in germ granules. Given the orientation of *nos* mRNA within germ granules, can the authors speculate how PABP specifically might affect its translation there, as it interacts with the 3' embedded region of the RNA? It would seem that the usual model of PABP stimulating translation by promoting RNA circularization and re-initiation would not work in this context.

5. This is perhaps a bit picky, but the paper (for instance line 26) refers to the 5' end of *nos* RNA being on or on the outside of the germ granule. While this is almost certainly true, the data however only show that the ORF is in that location. Perhaps this description can be modified.

I also caught two small presentational errors:

Line 306, remove 'the' before 'Oskar's'

Line 390, should read 'mRNAs' not 'mRNA'

G. References

The reference list is appropriate.

H. Clarity and context

The paper is extremely clear and well-written, and appropriate context is provided.

Reviewer #2:

Remarks to the Author:

In the manuscript "Repressor sequestration activates translation of germ granule localized mRNA" Chen et al examine the role of posterior germ granules in *Drosophila* embryos in regulating the translation of *nanos* mRNA. Specifically, here the authors dissect the contribution of germ granule protein binding, localization and repressor recruitment to localized *nanos* translation. Overall the manuscript is well written and provides insights into the mechanism of spatial translation regulation in

the *Drosophila* embryo. The study is important because it directly shows a role for a condensate in promoting the proportion of RNAs being translated and provides a clean mechanism for how a condensate can promote translation by localized relief of translation inhibition. This is an important contribution to understanding the function of condensates in the regulation of mRNA. A list of suggestions follows a brief summary of their findings.

To investigate translation at the single mRNA level the authors express a modified nanos mRNA with a SunTag array appended to it. This construct shows biased enrichment of scFV signal in the germlasm compared to the soma. Locally enriched scFV signal requires the presence of the germ granule protein Oskar. Nanos protein itself has been reported to form a posterior-to-anterior gradient, and the scFV localization reproduces this spatial bias. The authors suggest that bright scFV-GFP signal associated with suntag-nanos mRNAs represent active translation sites in polysomes (also because they are sensitive to puromycin) whereas dimmer spots represent previously translated single suntag-nanos proteins. They further suggest that the low expression level of suntag-nanos mRNA reduces the number of RNAs in germ granules and therefore rule out the possibility of these bright foci representing translation events occurring on multiple mRNAs. For the rest of the paper the authors interpret these polysome foci as representing active translation events.

Having established this translation visualization system, the authors next directly assess the impact of germ granules on the localization of nanos mRNA translation. They observe that translation is lost in the absence of germ granules (*oskar* KD) and further, ectopic anterior expression of *oskar* leads to corresponding anterior scFV signal. This shows that *oskar* protein and nanos mRNA interactions are sufficient to drive local translation.

The authors then study the localization and orientation of translation mRNAs relative to the boundary of germ granules. Translating mRNAs are enriched on granule surfaces whereas non-translating mRNAs appear enriched within germ granules. The authors use RPS6 staining to visualize ribosomes enrichment, and observe they are enriched at germ granules, and depleted outside them. These pieces of evidence point to local enrichment of nanos translation at the surface of, but not embedded within germ granules.

To study the mechanism of germ-granule associated translation, the authors first investigate the impact of Smaug binding by mutating the SRE element in nanos 3'UTR. They find that the level of translation does not vary in the soma and the germlasm, suggesting uniform translation in the absence of Smaug binding, and further interpret the uniform brightness of scFv stain as implying uniform initiation rates. To further understand the type of translation regulation by *smaug*, they measure ribosome elongation rates on SRE mutants and find them to be identical to that of WT 3'UTR. The authors conclude that germ granules enrich for translation by preventing its repression by Smaug, rather than promoting initiation or elongation rates.

They investigate the mechanism of how germ granules protect nanos mRNAs from Smaug repression by examining the physical localization of Smaug as well as the repressors of translation that it binds relative to germ granules. They observe that while Smaug itself is enriched within germ granules, translation repressors are not enriched in them, although repressor ME31B forms clusters on the surface of germ granules in pole cells. They conclude that Smaug may be prevented from being a translation repressor when sequestered into germ granules.

Finally the authors study repression of nanos translation in a strain where Oskar's interaction with

Smaug is disrupted. This mutant shows decreased nanos translation in germ granules but this decrease is similar in WT and SREmut 3'UTRs. The authors conclude that the oskar NQ mutant, by failing to bind and recruit Smaug to germ granules, also fails to counteract Smaug repression.

Specific suggestions:

1. The shifts in Extended Figure 5C-G can be made easier to understand by overlaying lines representing means of the distribution, either of the histograms or the KDE curves.
2. While the authors claim that the low expression level of suntag-nanos does not lead to homotypic assemblies, it appears from the images that the mRNA spots show variations in size and brightness. It would be interesting to see if there are correlations in the number/size of nanos RNA foci and the level of translation, to parse out whether homotypic RNA interactions in this case influence translation or simply germ granule localization and specificity.
3. The authors should consider discussing differences between the number of observed suntag-nanos mRNA per vasa granule (~1 from the images provided) vs previously reported numbers of nanos mRNA per vasa granule (~14 nanos molecules/granule; Trcek et al., Mol. Cell, 2019). This discussion point would be pertinent in the discussion section. Is this due to the appendage of the SunTag sequence? If so, what implications does this have for the native translation of nanos.
4. Line 197 – 200 it is worth dissecting the contribution of the SunTag sequence to the orientation of the suntag-nanos molecules within or on the surface of germ granules. Given the size and repetitiveness of the suntag sequence, it could be that this sequence contributes to the orientation of these fusion mRNAs in a way that doesn't faithfully report the orientation of native nanos mRNA. It would be good to check if smFISH directed at the 5' and 3' regions of native nanos mRNA recapitulate the orientation of translating suntag-nanos mRNA.
5. Lines 289 - 293. It is not clear from the images in Extended Data Fig. 11b, 12 that Smaug cofactors Cup, CCR4, NOT3 and ME31B are not enriched in germ granules. These extended data fig 11b and 12 would benefit from intensity profiles of the Smaug cofactor compared to the vasa signal, similar to those included in Figure 4e. In addition, while these factors may not be enriched in germ granules, they don't seem to be excluded (line scans could indicate something different - but from the current data shown, this seems to be the case). If this is the case, it isn't clear to me that Smaug-cofactor interactions are disrupted by the germ granule environment. I think this merits additional discussion acknowledging alternative mechanisms by which germ granules may activate translation of nanos.

Minor: In Extended data Figure 5c-f, use a y-axis with the same dimensions to allow for easier comparisons of distributions from the data to the controls used.

Reviewer #3:

Remarks to the Author:

Summary of the key results:

Chen et al. have utilized single molecule imaging to further elucidate the mechanism of nanos localization to germ granules and its impact on translation in the germline of *Drosophila*. The authors used the SunTag system to visualize endogenous translation of nanos and showed that translation colocalizes with germ granules. By visualizing both translation through SunTag and mRNA

with FISH, the authors found that translation occurs on the exterior border of germ granules, while translationally silent nanos transcripts were deeper within the germ granules. Through the use of FRAP, they found that the elongation rate of nanos translation in the germlasm was nearly equivalent to that of nanos transcripts in the soma, harboring a mutated 3'UTR to abolish translation repression. Although germ granules do not enhance translation dynamics of nanos, the authors found that germ granules promote nanos translation by sequestering the translation repressor protein Smaug using a novel germ granule scaffold mutant, Osk-NQmut. Thus, while nanos transcripts are localized to germ granules containing the translation repressor Smaug, translation is derepressed, indicating a higher order organization of germ granules. Overall the paper and methodology were well organized, the story clear, and the data well presented.

Originality and significance:

Given the growing research interest of RNPs and its associations with translational status of localized mRNAs, this study provides great insight into how mRNA localization impacts its translation in germ granules. Further this study illustrates a novel mechanism for the depression of nanos. However, some points should be addressed to further strengthen the claims proposed by the author:

Major points:

1. While the authors reported the localization of both translating and non-translating nanos to germ granules, there were no dynamic measurements of nanos and germ granule diffusion. The authors should quantify the co-movement of nanos and germ granules to confirm true localization of the transcript to the germ granules with fast imaging to complement Fig2. This would also show any potential transient movements of nanos in and out of germ granules. This is also relevant in Fig3G, where it appears as though a translation site hops to a different germ granule between the 160s and 260s time points. If these interactions are transient in nature, it could change the interpretation of the mechanism of derepression.
2. The authors should resolve the issue of the algorithm measuring transcript distance to the germ granules, as shown in Fig2B,D,F-H, and SFig5C-G. The loss of abundance at the granule border could be skewing the interpretation of the transcript positioning to the granule. The authors should also implement spatial cross correlation analysis to obtain a more precise quantification of the degree of positioning of transcripts at germ granules.
3. In SFig7A, some translation puncta were still apparent after puromycin treatment. This would suggest that there was incomplete polysome release, which could have potentially perturbed mRNA repositioning in the granule. The experiment should be repeated with a higher concentration of puromycin and/or comment on the non-responsive fraction. Alternative translation disruption drugs, such as the initiation inhibitor harringtonine, could also help demonstrate the impacts of mRNA positioning within the granule.

Minor points:

1. In SFig2D, translation was not visible at stage 5 with scFv but was visible with IF. This is confusing as the background does not appear that high in both images. To what degree do the mature protein accumulate and does this sequester the scFv?
2. In Fig3B, the percentage of translating SunTag-nanos-tub3'UTR was quantified for the soma but not

for the germlasm.

3. The authors conclude that the initiation rate SunTag-nanos-SREmut is the same in the germlasm and soma based on measuring translation intensity (Fig3C) before discussing elongation rate (Fig3E). However, this claim cannot be made without confirming that the elongation rate of this transcript is the same in the germlasm and soma. Thus, the conclusion about initiation rates should come after Fig. 3E.

4. In SFig5A, the inset image appears saturated. Since the dynamic range is so large, the image could be better displayed with a color gradient to visualize both bright and dim granules w/o saturation issues.

5. In SFig4E, it appears as though translation is occurring in cells depleted of Vasa. Are these cells containing translation void of germ granules?

6. The authors should consider citing/acknowledging a few papers demonstrating a similar analysis of mRNA interactions with granules in live cells using single-mRNA translation reporters:

>Moon and Morisaki paper showing translating mRNA can associate with granules and can be tethered so the 3' end transiently moves in and out of granules:

Moon SL, Morisaki T, Khong A, Lyon K, Parker R, Stasevich TJ. Multicolour single-molecule tracking of mRNA interactions with RNP granules. *Nat Cell Biol.* 2019 Feb;21(2):162-168. doi: 10.1038/s41556-018-0263-4. Epub 2019 Jan 21. PMID: 30664789; PMCID: PMC6375083.

>Adivarahan and Khong papers showing how positioning of 5' and 3' ends of mRNA can be measured within polysomes:

Adivarahan S, Livingston N, Nicholson B, Rahman S, Wu B, Rissland OS, Zenklusen D. Spatial Organization of Single mRNPs at Different Stages of the Gene Expression Pathway. *Mol Cell.* 2018 Nov 15;72(4):727-738.e5. doi: 10.1016/j.molcel.2018.10.010. Epub 2018 Nov 8. PMID: 30415950; PMCID: PMC6592633.

Khong A, Parker R. mRNP architecture in translating and stress conditions reveals an ordered pathway of mRNP compaction. *J Cell Biol.* 2018 Dec 3;217(12):4124-4140. doi: 10.1083/jcb.201806183. Epub 2018 Oct 15. PMID: 30322972; PMCID: PMC6279387.

READABILITY OF MANUSCRIPTS – Nature Cell Biology is read by cell biologists from diverse

backgrounds, many of whom are not native English speakers. Authors should aim to communicate their findings clearly, explaining technical jargon that might be unfamiliar to non-specialists, and avoiding non-standard abbreviations. Titles and abstracts should concisely communicate the main findings of the study, and the background, rationale, results and conclusions should be clearly explained in the manuscript in a manner accessible to a broad cell biology audience. Nature Cell Biology uses British spelling.

REFERENCES – are limited to a total of 70 for Articles, Resources, Technical Reports; and 40 for Letters. This includes references in the main text and Methods combined. References must be numbered sequentially as they appear in the main text, tables and figure legends and Methods and must follow the precise style of Nature Cell Biology references. References only cited in the Methods should be numbered consecutively following the last reference cited in the main text. References only associated with Supplementary Information (e.g. in supplementary legends) do not count toward the

total reference limit and do not need to be cited in numerical continuity with references in the main text. Only published papers can be cited, and each publication cited should be included in the numbered reference list, which should include the manuscript titles. Footnotes are not permitted.

Methods should be written concisely, but should contain all elements necessary to allow interpretation and replication of the results. As a guideline, Methods sections typically do not exceed 3,000 words. The Methods should be divided into subsections listing reagents and techniques. When citing previous methods, accurate references should be provided and any alterations should be noted. Information must be provided about: antibody dilutions, company names, catalogue numbers and clone numbers for monoclonal antibodies; sequences of RNAi and cDNA probes/primers or company names and catalogue numbers if reagents are commercial; cell line names, sources and information on cell line identity and authentication. Animal studies and experiments involving human subjects must be reported in detail, identifying the committees approving the protocols. For studies involving human subjects/samples, a statement must be included confirming that informed consent was obtained. Statistical analyses and information on the reproducibility of experimental results should be provided in a section titled "Statistics and Reproducibility".

All Nature Cell Biology manuscripts submitted on or after March 21 2016 must include a Data availability statement as a separate section after Methods but before references, under the heading "Data Availability". For Springer Nature policies on data availability see <http://www.nature.com/authors/policies/availability.html>; for more information on this particular policy see <http://www.nature.com/authors/policies/data/data-availability-statements-data-citations.pdf>. The Data availability statement should include:

- Accession codes for primary datasets (generated during the study under consideration and designated as "primary accessions") and secondary datasets (published datasets reanalysed during the study under consideration, designated as "referenced accessions"). For primary accessions data should be made public to coincide with publication of the manuscript. A list of data types for which submission to community-endorsed public repositories is mandated (including sequence, structure, microarray, deep sequencing data) can be found here <http://www.nature.com/authors/policies/availability.html#data>.
- Unique identifiers (accession codes, DOIs or other unique persistent identifier) and hyperlinks for datasets deposited in an approved repository, but for which data deposition is not mandated (see here for details <http://www.nature.com/sdata/data-policies/repositories>).
- At a minimum, please include a statement confirming that all relevant data are available from the authors, and/or are included with the manuscript (e.g. as source data or supplementary information), listing which data are included (e.g. by figure panels and data types) and mentioning any restrictions on availability.
- If a dataset has a Digital Object Identifier (DOI) as its unique identifier, we strongly encourage including this in the Reference list and citing the dataset in the Methods.

We recommend that you upload the step-by-step protocols used in this manuscript to the Protocol Exchange. More details can be found at www.nature.com/protocolexchange/about.

All imaging data should be accompanied by scale bars, which should be defined in the legend. Cropped images of gels/blots are acceptable, but need to be accompanied by size markers, and to retain visible background signal within the linear range (i.e. should not be saturated). The boundaries of panels with low background have to be demarked with black lines. Splicing of panels should only be considered if unavoidable, and must be clearly marked on the figure, and noted in the legend with a statement on whether the samples were obtained and processed simultaneously. Quantitative comparisons between samples on different gels/blots are discouraged; if this is unavoidable, it should only be performed for samples derived from the same experiment with gels/blots were processed in parallel, which needs to be stated in the legend.

- We do not recommend using Adobe Photoshop for designing figures, but we can accept Photoshop generated (.PSD or .TIFF) files only if each element included in the figure (text, labels, pictures, graphs, arrows and scale bars) are on separate layers. All text should be editable in 'type layers' and line-art such as graphs and other simple schematics should be preserved and embedded within 'vector

smart objects' - not flattened raster/bitmap graphics.

The total number of Supplementary Figures (not including the “unprocessed scans” Supplementary Figure) should not exceed the number of main display items (figures and/or tables (see our Guide to Authors and March 2012 editorial <http://www.nature.com/ncb/authors/submit/index.html#supinfo>; <http://www.nature.com/ncb/journal/v14/n3/index.html#ed>). No restrictions apply to Supplementary Tables or Videos, but we advise authors to be selective in including supplemental data.

GUIDELINES FOR EXPERIMENTAL AND STATISTICAL REPORTING

REPORTING REQUIREMENTS – We are trying to improve the quality of methods and statistics reporting in our papers. To that end, we are now asking authors to complete a reporting summary that collects information on experimental design and reagents. The Reporting Summary can be found here <https://www.nature.com/documents/nr-reporting-summary.pdf> If you would like to reference the guidance text as you complete the template, please access these flattened versions at <http://www.nature.com/authors/policies/availability.html>.

We strongly recommend the presentation of source data for graphical and statistical analyses as a separate Supplementary Table, and request that source data for all independent repeats are provided when representative experiments of multiple independent repeats, or averages of two independent experiments are presented. This supplementary table should be in Excel format, with data for different figures provided as different sheets within a single Excel file. It should be labelled and numbered as one of the supplementary tables, titled “Statistics Source Data”, and mentioned in all relevant figure legends.

Author Rebuttal to Initial comments

Comments to Reviewers NCB-A52217-T

We thank the reviewers for insightful and helpful feedback. We have performed additional experiments and analysis to address the issues raised by the reviewers. Here we summarize the main data and analysis added to the manuscript:

- 1) We resolved the issue in our distance-measuring program that caused the 'dip' at the zero of all the histograms. We have re-analyzed our data and updated new graphs without this artifact. We explain the cause and our solution in detail in response to reviewer #1.
- 2) To test whether the *suntag* may affect RNA positioning, we performed smFISH on endogenous *nanos* mRNA using 5' and 3' specific probes, in embryos and oocytes. The results with endogenous wildtype *nanos* are consistent with our analysis with *suntag-nanos*. We explain our findings in detail in the response to the reviewer #2.
- 3) To further test how translation inhibition affects RNA positioning, we tested a different translation inhibitor, Harringtonine. As described previously using Puromycin, we observed a similar result with regard to RNA 5' positioning using Harringtonine. Details are provided in the response to the reviewer #3.
- 4) To demonstrate the association between polysomes and germ granules, we added a live-imaging movie of translation foci with their attached germ granules for 450 seconds and performed manual tracking to show their co-movement over a long period. Details are provided in the response to the reviewer #3.

To accommodate these new results in the manuscript, we split our original Figure 2 into two main figures (now Figures 2 and 3). We also analyzed and discussed how the clustering of endogenous *nanos* mRNA may affect translation and how a lack of clustering by *suntag-nanos* may affect our interpretation of the results.

We also decided to change the title of our manuscript to 'Direct Observation of Translational Activation by a Ribonucleoprotein Granule'. We believe that this title captures the major novelty and take-home message of the paper better than the previous title and will attract a broader readership.

Below we provide detailed point-by-point responses to the issues raised by the reviewers.

Reviewers' Comments:

Reviewer #1:

Remarks to the Author:

A. Summary of the key results

The manuscript by Chen et al uses quantitative single-molecule imaging to make a definitive case that germ granules are the sites of translation of nanos (nos) mRNA in the early Drosophila embryo. It further shows that nanos mRNA is oriented within germ granules such that its ORF is outside or on the surface of the granules, while its 3' UTR is embedded within. Surprisingly, Smaug (Smg), a well-established translational repressor of nos mRNA, also accumulates within germ granules. Using a novel separation-of-function allele of oskar (osk) in which a predicted disordered region is altered, the authors showed that sequestration of Smg is dependent on Osk, and in this mutant nos translation is greatly reduced. Within germ granules Smg does not act as a translational repressor, because necessary co-factors such as Cup and the CCR/NOT complex do not accumulate there, but outside of germ granules Smg is active. Thus Osk controls nos translation by enabling Smg to accumulate into a region of the germ granules where it is inactive.

B. Originality and significance

The experiments involve original technologies. In my view these results are highly significant, not just because they answer a long-standing question in the field, but also because it is probable that similar mechanisms function in other ribonucleoprotein particles (RNPs) that form condensates. Such RNPs are ubiquitous in evolution and they operate in numerous cellular and developmental processes.

C. Data and methodology

Overall, the experiments in this manuscript are carefully performed and well documented. Appropriate controls are included.

D. Statistics and uncertainties

The use of statistics is thorough and appropriate, with uncertainties well documented except for point 1 below.

E. Conclusions

The conclusions are robustly supported by the data.

F. Suggested improvements

Because of the high quality of the paper I have only a few relatively minor comments that I would like the authors to address in revision.

1. In the legend to Extended Data Fig. 5, it is noted that measurements around 0nm are underrepresented in all experiments, and that this likely results from an artefact caused by the design of the distance measuring program. Can this be further explained, and how do the authors know that additional artefacts do not further distort the data? Better yet, can it be fixed?

We thank the reviewers for pointing out the issue, which was also brought up by reviewer #3. We have addressed this issue by reanalyzing the data with a revised program. The differences between the two approaches are summarized below. Importantly, the conclusions were unaffected by the specific distance-measuring program.

The under-representation around 0nm (the granule border) was an artifact caused by the original distance measuring program. In the original analysis pipeline, we represented each voxel (42.5nm x 42.5nm x 150nm) of the granule borders with a single point in the center of the voxel. The continuous granule border was thus represented by a discrete set of points (a 'point cloud') leading to an artificially sparse population of points at the border of the granule. Then we measured the distances of foci signals (smFISH or scFv translation foci) to their closest points in the 'point cloud' to represent their distances to the granule border. Due to the sparsity of points used to represent a continuous surface, foci right inside the border voxels, which are supposed to be measured as 0 nm from the border, could be measured as being further away from the border because of their distance from the closest representing point in the 'point cloud'. These foci thus had a relatively longer distance measurements and fell into the bins further away

from the border of the granule (represented by 0nm in the histogram), leading to a 'dip' around '0' in the histogram.

We fixed this issue by increasing the density of points used to represent the surface of the granule. We used 27 (3^3) points, equally spaced within each voxel, to represent the voxels of the granule border. This increased number of points allows most foci that are right on the border of the granule to have distance measurements within the bins closest to 0nm, thus faithfully reflecting their locations. We used this new program to re-analyze all the image data and updated all the distance measurement graphs in our manuscript. This resolved the under-representations around 0nm. Below we show two examples, the distributions of a million random points and *suntag* mRNA smFISH foci within a small section of germlasm. As shown in these graphs, the updated graphs generated by the new analysis pipeline do not have the 'dip' around 0nm without changing the overall histogram distribution. With the updated program and resulting new graphs, we demonstrate that all the conclusions drawn from the original graphs still hold.

2. In the Supplementary Notes, it is stated that *suntag-nanos* homozygotes are viable and females lay normal numbers of eggs, but the eggs do not hatch. The explanation given is that this phenotype results from low expression of *suntag-nanos* as compared with the endogenous gene. The phenotype of the embryos produced in the eggs should be described. Do they have posterior patterning defects consistent with a *nos* hypomorph, or is their phenotype different and/or more complex?

To assess whether SunTag-Nanos protein is active, we examined the abdominal segmentation phenotype of the embryos derived from mothers homozygous for the endogenously tagged *suntag-nanos* gene using cuticle preparations. As shown in the table below, embryos from heterozygous flies (with one copy of wildtype *nanos*) showed mostly normal abdominal segmentation. In contrast, embryos from homozygous *suntag-nanos* flies formed either no abdominal segments or 1-3 segments, which suggests a low Nanos activity generated by *suntag-nanos* mRNA. This low activity is likely a combinatorial result of low mRNA expression and perturbed protein function by the SunTag. We would like to note that Nanos is also required for the maintenance of germline stem cells (GSCs) in female flies, where it plays a role for germline stem cell maintenance, a prerequisite for oogenesis and egg laying. Notably, homozygous *suntag-nanos* female flies lay eggs normally, suggesting SunTag-Nanos functional to support the maintenance of germline stem cells. Thus, we consider *suntag-nanos* a hypomorphic mutation of *nanos*.

genotype	abdominal segment number				total n
	0	1 to 3	4 to 6	7 or 8	
suntag-nanos/TM3, Sb	0	0	3	97	100
suntag-nanos/suntag-nanos	56	38	1	0	95

3. Also relevant to the Supplementary Notes, given the failure to recover one, can the authors speculate as to whether a knock-in with the insertion as designed might cause dominant lethality or dominant sterility?

As we discussed above, the *suntag-nanos* gene we recovered from CRISPR mutagenesis has reduced, hypomorphic function. Heterozygous *suntag-nanos/wt* flies have normal fertility so this specific *suntag-nanos* allele does not cause dominant lethality or sterility. Curiously, we only recovered this one, low-expression allele, which also contained an off-target disruption of the *nanos* promoter. This suggests that at normal expression level (comparable to endogenous *nanos*) *suntag-nanos* may become toxic to the germline and may cause dominant sterility, which would have precluded recovery of the line.

Alternative explanation to the low success rate of generating *suntag-nanos* knock-in is that CRISPR-Cas9 mediated cutting at the *nanos* locus is inefficient. In fact, attempts at knocking in other tags at the *nanos* locus in our lab have also been largely unsuccessful.

4. In lines 231-232 there is speculation that recruitment of translation factors to germ granules may facilitate mRNA translation and complement the derepression mechanism that is the major focus of the paper. Extended Data Fig. 9 indeed shows that PABP and eIF4G are enriched in germ granules. Given the orientation of nos mRNA within germ granules, can the authors speculate how PABP specifically might affect its translation there, as it interacts with the 3' embedded region of the RNA? It would seem that the usual model of PABP stimulating translation by promoting RNA circularization and re-initiation would not work in this context.

We observed that PABP tends to form foci that are associated with germ granules and that the association is heterogeneous. Namely, some granules associate with large PABP foci while some have small ones or no association. Additionally, PABP foci did not tend to localize to the center of germ granules, where most polyA tails are likely located. This suggests that the localization or association of PABP with germ granules may not be due to PABP-polyA tail interaction. Instead, PABP may localize through interaction with germ granule proteins and form foci through homotypic clustering or phase separation. PABP localized in this way may still stimulate translation initiation on the granule surface by interacting with eIF4G and stabilizing the eIF4F complex, although the stimulation may not be specific to specific mRNAs.

Our model of mRNA orientation on germ granules suggests that translating mRNA molecules have an extended conformation while untranslated mRNA molecules have a circularized conformation. We recognize that this model suggests the opposite of the usual circularization model of translation initiation. However, it is supported by more recent studies that, like us, observe an extended conformation of translating mRNA (Adivarahan et al., 2018; Khong and Parker, 2018). Therefore, the conformation of translating mRNA can be context-dependent or specific to different mRNAs. We have added this observation to the discussion of the revised manuscript.

5. This is perhaps a bit picky, but the paper (for instance line 26) refers to the 5' end of nos RNA being on or on the outside of the germ granule. While this is almost certainly true, the data however only show that the ORF is in that location. Perhaps this description can be modified.

We shared the concerns of the reviewer. We therefore used smFISH probes against the native *nanos* 5' sequence (including the 5'UTR and 415nt of the ORF next to the 5'UTR) and probes against *nanos* 3'UTR to visualize native *nanos* mRNA in germlasm, in a translationally silent state during oogenesis and a

translationally active state in the early embryo. We find that the *nanos* 5' probe smFISH signal coats around the 3' signal and the surface of embryonic germ granules, while the probes overlap in germ granules during oogenesis. This is consistent with our model that the coding sequence and 5' UTR of *nanos* is exposed on the surface of the germ granules while the 3'UTR is anchored inside the granules. Response #4 to reviewer #2 comments this point in more detail.

I also caught two small presentational errors:

Line 306, remove 'the' before 'Oskar's'

Line 390, should read 'mRNAs' not 'mRNA'

We thank the reviewer for pointing these out. We have made changes in the revised manuscript.

G. References

The reference list is appropriate.

H. Clarity and context

The paper is extremely clear and well-written, and appropriate context is provided.

Reviewer #2:

Remarks to the Author:

*In the manuscript "Repressor sequestration activates translation of germ granule localized mRNA" Chen et al examine the role of posterior germ granules in *Drosophila* embryos in regulating the translation of *nanos* mRNA. Specifically, here the authors dissect the contribution of germ granule protein binding, localization and repressor recruitment to localized *nanos* translation. Overall the manuscript is well written and provides insights into the mechanism of spatial translation regulation in the *Drosophila* embryo. The study is important because it directly shows a role for a condensate in promoting the proportion of RNAs being translated and provides a clean mechanism for how a condensate can promote translation by localized relief of translation inhibition. This is an important contribution to understanding the function of condensates in the regulation of mRNA. A list of suggestions follows a brief summary of their findings.*

*To investigate translation at the single mRNA level the authors express a modified *nanos* mRNA with a*

SunTag array appended to it. This construct shows biased enrichment of scFV signal in the germlasm compared to the soma. Locally enriched scFV signal requires the presence of the germ granule protein Oskar. Nanos protein itself has been reported to form a posterior-to-anterior gradient, and the scFV localization reproduces this spatial bias. The authors suggest that bright scFV-GFP signal associated with suntag-nanos mRNAs represent active translation sites in polysomes (also because they are sensitive to puromycin) whereas dimmer spots represent previously translated single suntag-nanos proteins. They further suggest that the low expression level of suntag-nanos mRNA reduces the number of RNAs in germ granules and therefore rule out the possibility of these bright foci representing translation events occurring on multiple mRNAs. For the rest of the paper the authors interpret these polysome foci as representing active translation events.

Having established this translation visualization system, the authors next directly assess the impact of germ granules on the localization of nanos mRNA translation. They observe that translation is lost in the absence of germ granules (oskar KD) and further, ectopic anterior expression of oskar leads to corresponding anterior scFV signal. This shows that oskar protein and nanos mRNA interactions are sufficient to drive local translation.

The authors then study the localization and orientation of translation mRNAs relative to the boundary of germ granules. Translating mRNAs are enriched on granule surfaces whereas non-translating mRNAs appear enriched within germ granules. The authors use RPS6 staining to visualize ribosomes enrichment, and observe they are enriched at germ granules, and depleted outside them. These pieces of evidence point to local enrichment of nanos translation at the surface of, but not embedded within germ granules.

To study the mechanism of germ-granule associated translation, the authors first investigate the impact of Smaug binding by mutating the SRE element in nanos 3'UTR. They find that the level of translation does not vary in the soma and the germlasm, suggesting uniform translation in the absence of Smaug binding, and further interpret the uniform brightness of scFv stain as implying uniform initiation rates. To further understand the type of translation regulation by smaug, they measure ribosome elongation rates on SRE mutants and find them to be identical to that of WT 3'UTR. The authors conclude that germ granules enrich for translation by preventing its repression by Smaug, rather than promoting initiation or elongation rates.

They investigate the mechanism of how germ granules protect nanos mRNAs from Smaug repression by examining the physical localization of Smaug as well as the repressors of translation that it binds relative to germ granules. They observe that while Smaug itself is enriched within germ granules, translation repressors are not enriched in them, although repressor ME31B forms clusters on the surface of germ granules in pole cells. They conclude that Smaug may be prevented from being a translation repressor when sequestered into germ granules.

Finally the authors study repression of *nanos* translation in a strain where *Oskar*'s interaction with *Smaug* is disrupted. This mutant shows decreased *nanos* translation in germ granules but this decrease is similar in WT and SREmut 3'UTRs. The authors conclude that the *oskar* NQ mutant, by failing to bind and recruit *Smaug* to germ granules, also fails to counteract *Smaug* repression.

Specific suggestions:

1. The shifts in Extended Figure 5C-G can be made easier to understand by overlaying lines representing means of the distribution, either of the histograms or the KDE curves.

We added ticks to the Extended figure 5g to represent the means of each distribution.

2. While the authors claim that the low expression level of *suntag-nanos* does not lead to homotypic assemblies, it appears from the images that the mRNA spots show variations in size and brightness. It would be interesting to see if there are correlations in the number/size of *nanos* RNA foci and the level of translation, to parse out whether homotypic RNA interactions in this case influence translation or simply germ granule localization and specificity.

The reviewer raises an important point that not all *suntag-nanos* mRNA smFISH foci in our image have only one RNA and that a cluster of multiple mRNAs may affect the quantification and interpretation and possibly hint to a role for homotypic mRNA clusters for translational efficiency. We thus analyzed the degree of homotypic RNA clustering in our images. By measuring the intensities of smFISH foci and normalizing them with single mRNA intensity (measured using the smFISH foci signal in soma, which has only a single mRNA), we derived the copy numbers of RNA in individual foci in germlasm. As shown in the figure below (left panel), around 90% of foci have only one *suntag* mRNA, 5-10% of foci have two, and even fewer foci have more than two. Therefore, the degree of clustering is rather low and most of our quantification and analysis were performed on single mRNAs.

We also agree with the reviewer that it is an interesting question whether RNA clustering can affect the level of translation. We analyzed the percentage of being translation-positive (middle panel) and the scFv-GFP foci intensity (right panel) of the smFISH foci with one, two, or three mRNAs. If the clustering can facilitate translation efficiency, the percentage or the intensity should more than double or triple with two or three mRNAs in a cluster, which was not what we observed. A more rigorous study on how clustering affects translation requires a mathematical model to fit our data and a stronger expression of *suntag-nanos*, which allows the analysis of the cluster with a higher amount of mRNA. We have added some of these consideration to the discussion in the supplementary text.

3. The authors should consider discussing differences between the number of observed *suntag-nanos* mRNA per vasa granule (~1 from the images provided) vs previously reported numbers of *nanos* mRNA per vasa granule (~14 *nanos* molecules/granule; Trcek et al., Mol. Cell, 2019). This discussion point would be pertinent in the discussion section. Is this due to the appendage of the SunTag sequence? If so, what implications does this have for the native translation of *nanos*.

This comment is related to comment #2. Homotypic clustering is dependent on the abundance of specific mRNA (Trcek et al., 2020). The lack of clustering of *suntag-nanos* is due to its low expression. Indeed, we could increase the clustering of *suntag-nanos* when we increased its expression with the UAS-GAL4 system. Our preliminary analysis of *suntag-nanos* clusters described above did not suggest an enhancing effect of clustering on translation but more rigorous and systematic studies are required to address this question.

4. Line 197 – 200 it is worth dissecting the contribution of the SunTag sequence to the orientation of the *suntag-nanos* molecules within or on the surface of germ granules. Given the size and repetitiveness of the *suntag* sequence, it could be that this sequence contributes to the orientation of these fusion mRNAs in a way that doesn't faithfully report the orientation of native *nanos* mRNA. It would be good to check if smFISH directed at the 5' and 3' regions of native *nanos* mRNA recapitulate the orientation of translating *suntag-nanos* mRNA.

To address the reviewer's concern that the *suntag* sequence may have created an artifact for mRNA positioning, we used smFISH probes against 5' UTR (5' UTR plus part of the ORF next to 5' UTR) and 3' UTR

of *nanos* to visualize native *nanos* mRNA. The results have been included in the revised manuscript fig 2g-2i. Basically, we found that the *nanos* 5' probe smFISH signal 'coated' the 3' probe signal and was associated with the surface of germ granules (Oskar-GFP as marker). The 3' probe smFISH signal was colocalized with germ granule proteins, compared to the 5' probe as quantified using Pearson Correlation Coefficient (PCC). These results are consistent with our observation with *suntag-nanos* mRNA and support the conclusion that the 5' end and coding sequence of translating mRNAs tend to orient toward the outside of germ granules while the 3'UTRs are anchored inside.

5. Lines 289 - 293. It is not clear from the images in Extended Data Fig. 11b, 12 that *Smaug* cofactors *Cup*, *CCR4*, *NOT3* and *ME31B* are not enriched in germ granules. These extended data fig 11b and 12 would benefit from intensity profiles of the *Smaug* cofactor compared to the *vasa* signal, similar to those included in Figure 4e. In addition, while these factors may not be enriched in germ granules, they don't seem to be excluded (line scans could indicate something different - but from the current data shown, this seems to be the case). If this is the case, it isn't clear to me that *Smaug*-cofactor interactions are disrupted by the

germ granule environment. I think this merits additional discussion acknowledging alternative mechanisms by which germ granules may activate the translation of nanos.

We followed the reviewer's suggestion and added line profiles showing the distribution of each Smaug co-factor (ME31B, Cup, CCR4, NOT3) relative to germ granules. These proteins are neither enriched nor excluded from the germ granule environment. Considering that Smaug forms a stable repression complex with its co-factors and its target mRNA (Jeske et. al., 2010), the observed absence of enrichment of these co-factors with Smaug in germ granules leads us to speculate that the interaction may be disrupted, allowing translation to occur. Alternatively, translational de-repression and the absence of enrichment of these co-factors could also indicate that that Smaug may not be able to access target *nanos* mRNA in germ granules, which can be a prerequisite for the repression complex to assemble. In the discussion section, we included this alternative mechanism: *"Alternatively, interactions with germ granule proteins can constrain the mobility of Smaug and thus limit its access to the target mRNA nanos"*.

Minor: In Extended data Figure 5c-f, use a y-axis with the same dimensions to allow for easier comparisons of distributions from the data to the controls used.

We thank the reviewer for this suggestion and have made the dimensions of all y-axes equal.

Reviewer #3:

Remarks to the Author:

Summary of the key results:

Chen et al. have utilized single molecule imaging to further elucidate the mechanism of nanos localization to germ granules and its impact on translation in the germlasm of Drosophila. The authors used the SunTag system to visualize endogenous translation of nanos and showed that translation colocalizes with germ granules. By visualizing both translation through SunTag and mRNA with FISH, the authors found that translation occurs on the exterior border of germ granules, while translationally silent nanos transcripts were deeper within the germ granules. Through the use of FRAP, they found that the elongation rate of nanos translation in the germlasm was nearly equivalent to that of nanos transcripts in the soma, harboring a mutated 3'UTR to abolish translation repression. Although germ granules do not enhance translation dynamics of nanos, the authors found that germ granules promote nanos translation by sequestering the translation repressor protein Smaug using a novel germ granule scaffold mutant, Osk-NQmut. Thus, while nanos transcripts are localized to germ granules containing the translation repressor Smaug, translation is derepressed, indicating a higher order organization of germ granules. Overall the paper and methodology were well organized, the story clear, and the data well

presented.

Originality and significance:

Given the growing research interest of RNPs and its associations with translational status of localized mRNAs, this study provides great insight into how mRNA localization impacts its translation in germ granules. Further this study illustrates a novel mechanism for the depression of nanos. However, some points should be addressed to further strengthen the claims proposed by the author:

Major points:

1. While the authors reported the localization of both translating and non-translating nanos to germ granules, there were no dynamic measurements of nanos and germ granule diffusion. The authors should quantify the co-movement of nanos and germ granules to confirm true localization of the transcript to the germ granules with fast imaging to complement Fig2. This would also show any potential transient movements of nanos in and out of germ granules. This is also relevant in Fig3G, where it appears as though a translation site hops to a different germ granule between the 160s and 260s time points. If these interactions are transient in nature, it could change the interpretation of the mechanism of derepression.

The stable co-movement of *nanos* mRNA with germ granules using *nanos-MS2* has been demonstrated previously (Lerit and Gavis, 2011). The mobility of *nanos* mRNA associated with germ granules has also been quantified with FRAP experiment with *nanos-MS2* (Trcek et. al., 2020), which showed that over 70% of *nanos* mRNA molecules were immobile and that the mobile fraction had a 150s moving half-time (about 30s for Vasa protein). This result suggested that most of the *nanos* mRNA in germlasm associated with germ granules stably and that the mobile fraction had a slow exchange rate among granules. Therefore, we assume that *nanos* mRNA moving between granules to be a rare event.

While we have made *suntag-nanos-MS2* transgenic flies allowing simultaneous imaging of mRNA, translation, and germ granules, a description of this work is beyond the scope of the paper and we would like to reserve the data for another publication. However, to address the comment by the reviewer, we performed live imaging of germ granules and translation spots (fig 2a, supplemental movie 2). We manually tracked and recorded the co-movement of two translation spots and their associated germ granules (fig 2b). We observed that the translation spots were associated with the same granules throughout the live imaging, despite seeing dynamic and transient changes in granule morphology and the relative position of translation spots.

2. The authors should resolve the issue of the algorithm measuring transcript distance to the germ granules, as shown in Fig2B,D,F-H, and SFig5C-G. The loss of abundance at the granule border could be skewing the interpretation of the transcript positioning to the granule. The authors should also implement spatial cross correlation analysis to obtain a more precise quantification of the degree of positioning of transcripts at germ granules.

A similar concern was also raised by reviewer 1 and has been addressed in detail in our response to comment #1 of reviewer 1. Briefly, by changing the distance measuring program, we were able to resolve the 'dip' around the zero/border. All distance measurements were replaced with the new algorithm and all graphs updated. This change in methodology did not affect the overall conclusions drawn from our experiments.

3. In SFig7A, some translation puncta were still apparent after puromycin treatment. This would suggest that there was incomplete polysome release, which could have potentially perturbed mRNA repositioning in the granule. The experiment should be repeated with a higher concentration of puromycin and/or comment on the non-responsive fraction. Alternative translation disruption drugs, such as the initiation inhibitor harringtonine, could also help demonstrate the impacts of mRNA positioning within the granule.

Indeed, we observed that 15-30min after puromycin injection, there were always residual translation signals associated with mRNA (see the image below). We injected puromycin at 10mg/ml, which is 1000-fold of its normal working concentration. Therefore, the presence of the non-responsive fraction is unlikely due to insufficient puromycin concentration.

We tested harringtonine as an alternative as per the reviewer's advice. While puromycin requires micro-injection, we found that harringtonine can easily infiltrate embryos that are permeabilized by heptane, allowing quick and easy application of the drug. Additionally, injection at the germplasm can introduce artificial disruption in germ granule morphology, which is prevented using the permeabilization method. Therefore, harringtonine treatment is a better inhibitor to use for analyzing the effect on mRNA positioning. Indeed, the translation signal was completely abolished after harringtonine treatment. Nevertheless, we found that the effect on mRNA 5' position was minor and statistically insignificant after harringtonine treatment. We concluded that translational inhibition by drugs that interfere with translational initiation do not significantly affect the orientation of the 5' end of mRNA on germ granules.

Minor

points:

1. In *SFig2D*, translation was not visible at stage 5 with scFv but was visible with IF. This is confusing as the background does not appear that high in both images. To what degree do the mature protein accumulate and does this sequester the scFv?

The mature protein accumulates over time and appears as the dim spots in the background. These spots are not restricted to the germplasm but diffuse away into the somatic region, which is why there is no high background in the pole cells of stage 5. We believe that the accumulated mature protein depleted the limiting scFv and abolished the translation signal in pole cells. We have an accompanying preprint explaining this depletion in detail (Bellec et. al., 2023. bioRxiv 2023.02.25.529998)

2. In Fig3B, the percentage of translating SunTag-nanos-tub3'UTR was quantified for the soma but not for the germlasm.

We updated the graph and now include quantification of the translation in soma and germlasm.

3. The authors conclude that the initiation rate SunTag-nanos-SREmut is the same in the germlasm and soma based on measuring translation intensity (Fig3C) before discussing elongation rate (Fig3E). However, this claim cannot be made without confirming that the elongation rate of this transcript is the same in the germlasm and soma. Thus, the conclusion about initiation rates should come after Fig. 3E.

We thank the reviewer for pointing this out. In the new manuscript, we make the conclusion about the initiation rate after the conclusion of the same elongation rate.

4. In SFig5A, the inset image appears saturated. Since the dynamic range is so large, the image could be better displayed with a color gradient to visualize both bright and dim granules w/o saturation issues.

We thank the reviewer for this comment and like to clarify the purpose of this figure. In this image we want to illustrate how we used segmentation of germ granules to delineate a border for membraneless granules. The image is not used to illustrate quantification of intensities, so saturation should not be an issue. Also, we wanted to make the color tones similar between the image and the segmentation (black and white) so that readers can compare more easily. Therefore, we decided to keep the image in greyscale.

5. In SFig4E, it appears as though translation is occurring in cells depleted of Vasa. Are these cells containing translation void of germ granules?

In these low-mag images, the Vasa signals are relatively low to capture the granules at the periphery of the germlasm, where the Vasa signal in granules is significantly weaker than for granules in the center of the germlasm. In the revised manuscript, we adjusted the threshold of the Vasa channel so that the granules at the periphery are clearly visible.

6. The authors should consider citing/acknowledging a few papers demonstrating a similar analysis of mRNA interactions with granules in live cells using single-mRNA translation reporters:

>Moon and Morisaki paper showing translating mRNA can associate with granules and can be tethered so the 3' end transiently moves in and out of granules:

Moon SL, Morisaki T, Khong A, Lyon K, Parker R, Stasevich TJ. Multicolour single-molecule tracking of mRNA interactions with RNP granules. *Nat Cell Biol.* 2019 Feb;21(2):162-168. doi: 10.1038/s41556-018-0263-4. Epub 2019 Jan 21. PMID: 30664789; PMCID: PMC6375083.

>Adivarahan and Khong papers showing how positioning of 5' and 3' ends of mRNA can be measured within polysomes:

Adivarahan S, Livingston N, Nicholson B, Rahman S, Wu B, Rissland OS, Zenklusen D. Spatial Organization of Single mRNPs at Different Stages of the Gene Expression Pathway. *Mol Cell.* 2018 Nov 15;72(4):727-738.e5. doi: 10.1016/j.molcel.2018.10.010. Epub 2018 Nov 8. PMID: 30415950; PMCID: PMC6592633.

Khong A, Parker R. mRNP architecture in translating and stress conditions reveals an ordered pathway of mRNP compaction. *J Cell Biol.* 2018 Dec 3;217(12):4124-4140. doi: 10.1083/jcb.201806183. Epub 2018 Oct 15. PMID: 30322972; PMCID: PMC6279387.

We thank the reviewer for suggesting these references. We have cited Adivarahan et al., and Khong and Parker in our revised manuscript.

Decision Letter, first revision:

Our ref: NCB-A52217A

3rd April 2024

Dear Dr. Lehmann,

Thank you for submitting your revised manuscript "Direct Observation of Translational Activation by a Ribonucleoprotein Granule" (NCB-A52217A). It has now been seen by the original referees and their comments are below. The reviewers find that the paper has improved in revision, and therefore we'll be happy in principle to publish it in Nature Cell Biology, pending minor revisions to satisfy the referees' final requests and to comply with our editorial and formatting guidelines.

Thank you again for your interest in Nature Cell Biology Please do not hesitate to contact me if you have any questions.

Sincerely,
Daryl

Daryl Jason Verzosa David, PhD

Senior Editor, Nature Cell Biology
Nature Portfolio
Advisory Editor, npj Biological Physics and Mechanics

Heidelberger Platz 3, 14197 Berlin, Germany
Email: daryl.david@nature.com
ORCID: <https://orcid.org/0000-0002-9253-4805>

Reviewer #1 (Remarks to the Author):

The concerns I expressed in my previous review have all been satisfactorily and thoroughly addressed, most importantly with the improvements to the distance-measuring program that have eliminated an artefact that existed in the earlier version of the paper. I therefore recommend acceptance of this

manuscript for publication.

Paul Lasko (reviewer 1)

Reviewer #2 (Remarks to the Author):

I am satisfied with this revised version and happy to support acceptance of this manuscript.

Reviewer #3 (Remarks to the Author):

The authors have addressed all of my concerns.

Decision Letter, final checks:

Our ref: NCB-A52217A

19th April 2024

Dear Dr. Lehmann,

Thank you for your patience as we've prepared the guidelines for final submission of your Nature Cell Biology manuscript, "Direct Observation of Translational Activation by a Ribonucleoprotein Granule" (NCB-A52217A). Please carefully follow the step-by-step instructions provided in the attached file, and add a response in each row of the table to indicate the changes that you have made. Ensuring that each point is addressed will help to ensure that your revised manuscript can be swiftly handed over to our production team.

In recognition of the time and expertise our reviewers provide to Nature Cell Biology's editorial process, we would like to formally acknowledge their contribution to the external peer review of your manuscript entitled "Direct Observation of Translational Activation by a Ribonucleoprotein Granule". For those reviewers who give their assent, we will be publishing their names alongside the published article.

Nature Cell Biology offers a Transparent Peer Review option for new original research manuscripts submitted after December 1st, 2019. As part of this initiative, we encourage our authors to support increased transparency into the peer review process by agreeing to have the reviewer comments, author rebuttal letters, and editorial decision letters published as a Supplementary item. When you submit your final files please clearly state in your cover letter whether or not you would like to participate in this initiative. Please note that failure to state your preference will result in delays in accepting your manuscript for publication.

Cover suggestions

COVER ARTWORK: We welcome submissions of artwork for consideration for our cover. For more information, please see our guide for cover artwork.

Nature Cell Biology has now transitioned to a unified Rights Collection system which will allow our Author Services team to quickly and easily collect the rights and permissions required to publish your work. Approximately 10 days after your paper is formally accepted, you will receive an email in providing you with a link to complete the grant of rights. If your paper is eligible for Open Access, our Author Services team will also be in touch regarding any additional information that may be required to arrange payment for your article.

Please note that *Nature Cell Biology* is a Transformative Journal (TJ). Authors may publish their research with us through the traditional subscription access route or make their paper immediately open access through payment of an article-processing charge (APC). Authors will not be required to make a final decision about access to their article until it has been accepted. Find out more about Transformative Journals

Please use the following link for uploading these materials:
[Redacted]

Best regards,

Kendra Donahue
Staff
Nature Cell Biology

On behalf of

Daryl Jason Verzosa David, PhD

Senior Editor, Nature Cell Biology
Nature Portfolio
Advisory Editor, npj Biological Physics and Mechanics

Heidelberger Platz 3, 14197 Berlin, Germany
Email: daryl.david@nature.com
ORCID: <https://orcid.org/0000-0002-9253-4805>

Reviewer #1:

Remarks to the Author:

The concerns I expressed in my previous review have all been satisfactorily and thoroughly addressed, most importantly with the improvements to the distance-measuring program that have eliminated an artefact that existed in the earlier version of the paper. I therefore recommend acceptance of this manuscript for publication.

Paul Lasko (reviewer 1)

Reviewer #2:

Remarks to the Author:

I am satisfied with this revised version and happy to support acceptance of this manuscript.

Reviewer #3:

Remarks to the Author:

The authors have addressed all of my concerns.

Author Rebuttal, first revision:

Response to Reviewers NCB-A52217-B

The reviewers had no further comments, so we thank them again for constructive feedback which helped us to improve the manuscript.

Final Decision Letter:

Dear Dr Lehmann,

I am pleased to inform you that your manuscript, "Direct Observation of Translational Activation by a Ribonucleoprotein Granule", has now been accepted for publication in Nature Cell Biology.

Please note that *Nature Cell Biology* is a Transformative Journal (TJ). Authors may publish their research with us through the traditional subscription access route or make their paper immediately open access through payment of an article-processing charge (APC). Authors will not be required to make a final decision about access to their article until it has been accepted. Find out more about Transformative Journals

If you have not already done so, we strongly recommend that you upload the step-by-step protocols used in this manuscript to protocols.io (<https://protocols.io>), an open online resource that allows researchers to share their detailed experimental know-how. All uploaded protocols are made freely available and are assigned DOIs for ease of citation. Protocols and Nature Portfolio journal papers in which they are used can be linked to one another, and this link is clearly and prominently visible in the online versions of both. Authors who performed the specific experiments can act as primary authors for the Protocol as they will be best placed to share the methodology details, but the Corresponding Author of the present research paper should be included as one of the authors. By uploading your Protocols onto protocols.io, you are enabling researchers to more readily reproduce or adapt the methodology you use, as well as increasing the visibility of your protocols and papers. You can also establish a dedicated workspace to collect your lab Protocols. Further information can be found at <https://www.protocols.io/help/publish-articles>.

You can use a single sign-on for all your accounts, view the status of all your manuscript submissions and reviews, access usage statistics for your published articles and download a record of your

refereeing activity for the Nature Portfolio.

With kind regards,

Daryl

Daryl Jason Verzosa David, PhD

Senior Editor, Nature Cell Biology
Advisory Editor, npj Biological Physics and Mechanics
Nature Portfolio

Heidelberger Platz 3, 14197 Berlin, Germany
Email: daryl.david@nature.com
ORCID: <https://orcid.org/0000-0002-9253-4805>

** Visit the Springer Nature Editorial and Publishing website at www.springernature.com/editorial-and-publishing-jobs for more information about our career opportunities. If you have any questions please click here.**